



# Estimating Brewer-Dobson circulation trends from changes in stratospheric water vapour and methane

Liubov Poshyvailo-Strube[1,2,3], Rolf Müller[1], Stephan Fueglistaler[4,5], Michaela I. Hegglin[7], Johannes C. Laube[1], C. Michael Volk[6], and Felix Ploeger[1,6]

[1]Institute of Energy and Climate Research: Stratosphere (IEK-7), Forschungszentrum Jülich, Jülich, Germany
[2]Institute of Bio- and Geosciences: Agrosphere (IBG-3) Forschungszentrum Jülich, Jülich, Germany
[3]Centre for High-Performance Scientific Computing in Terrestrial Systems (HPSC TerrSys), Geoverbund ABC/J, Jülich, Germany
[4]Program in Atmospheric and Oceanic Sciences, Princeton University, Princeton, NJ, USA
[5]Department of Geosciences, Princeton University, Princeton, NJ, USA
[6]University of Wuppertal, Institute for Atmospheric and Environmental Research, Wuppertal, Germany
[7]Department of Meteorology, University of Reading, Reading, UK

**Correspondence:** Liubov Poshyvailo-Strube (l.poshyvailo@fz-juelich.de)

**Abstract.** The stratospheric meridional overturning circulation, also referred to as the Brewer-Dobson circulation (BDC), controls the composition of the stratosphere, which, in turn, affects radiation and climate. As the BDC cannot be directly measured, one has to infer its strength and trends indirectly. For instance, trace gas measurements allow the calculation of average transit times.

Satellite measurements provide information on the distributions of trace gases for the entire stratosphere, with measurements of particularly long and dense coverage available for stratospheric water vapour ($H_2O$). Although chemical processes and boundary conditions confound interpretation, the influence of $CH_4$ oxidation on $H_2O$ is relatively straightforward, and thus $H_2O$ is an appealing tracer for transport analysis despite these caveats. In this work, we explore how mean age of air trends can be estimated from the combination of stratospheric $H_2O$ and $CH_4$ data. We carry out different sensitivity studies with

the Chemical Lagrangian Model of the Stratosphere (CLaMS) and focus on the analysis of the periods of 1990-2006 and 1990-2017. In particular, we assess the methodological uncertainties related to the two commonly-used approximations of (i) instantaneous stratospheric entry mixing ratio propagation, and (ii) constant correlation between mean age and the fractional release factor of methane.

    Our results show that the estimated mean age of air trends from the combination of observed stratospheric $H_2O$ and $CH_4$

changes may be significantly affected by the assumed approximations. Depending on the investigated stratospheric region and the considered period, the error in estimated mean age of air decadal trends can be large – the discrepancies are up to 10 % per decade or even more at the lower stratosphere. For particular periods, the errors from the two approximations can lead to opposite effects, which may even cancel out. Finally, we propose an improvement to the approximation method by using an idealised age spectrum to propagate stratospheric entry mixing ratios. The findings of this work can be used for improving and

assessing the uncertainties in stratospheric BDC trend estimation from global satellite measurements.



# 1 Introduction

The stratospheric Brewer-Dobson circulation (BDC) affects the atmospheric distributions of radiatively active trace gases and is an important element in the climate system. This global-scale circulation transports air masses upwards in the tropics,

polewards, and downwards in middle and high latitudes (e.g., Holton et al., 1995). A particularly important greenhouse gas affected by the BDC is stratospheric water vapour ($H_2O$), which induces cooling of the stratosphere and warming of the troposphere (e.g. Solomon et al., 2010; Maycock et al., 2011; Riese et al., 2012). The reliability of climate model predictions is significantly affected by the representation of the processes controlling the distribution of stratospheric $H_2O$. Although the BDC is such a crucial factor influencing stratospheric $H_2O$ and Earth's climate, its long-term trends and associated effects on

transport and dynamics are not well understood. In particular, climate models predict a strengthening BDC in a future climate with increasing greenhouse gas levels (e.g., Butchart, 2014), whereas trace gas observations show only insignificant changes (Engel et al., 2017; Fritsch et al., 2020).

Because of the slowness of BDC transport (global transit times on the order of years) and its zonal mean character, direct measurements of related circulation velocities are not possible and the circulation must be inferred from temperature or trace

gas observations. A commonly used diagnostic for BDC transport is the mean age of air (AoA), the average time scale for transport through the stratosphere. Garcia et al. (2011) pointed out that the AoA is difficult to estimate from sparse stratospheric observations. However, under some conditions, it is possible to infer AoA from trace gas concentrations (e.g., Hall and Plumb, 1994; Strunk et al., 2000; Engel et al., 2009). Suitable species are so-called "clock tracers" – trace gases with a linearly increasing source – which can provide the first moment of the age spectrum, the mean age of air AoA (Waugh and Hall, 2002;

Schoeberl et al., 2005). Two examples of such tracers are $SF_6$ or $CO_2$, and these are frequently used to infer stratospheric AoA (e.g., Stiller et al., 2017; Engel et al., 2017). However, the availability of suitable observations with global coverage and extending over sufficiently long time periods necessary for estimating trends is very limited. Also, there are other complications, for instance $SF_6$ has a strong mesospheric sink, and $CO_2$ has seasonal cycle causing problems with inferring mean ages. Hence, several recent and ongoing research activities focus on trace gas species other than $SF_6$ and $CO_2$, to infer stratospheric AoA

and information on the BDC (e.g., Linz et al., 2017; Leedham Elvidge et al., 2018).

In particular, trace gas species with chemical sinks in the stratosphere provide information on the stratospheric circulation as the transport through the sink regions depends on the strength and depth of the circulation. For such long-lived species with stratospheric sinks, like $N_2O$, $CH_4$, or the chlorofluorocarbons (CFCs), the chemical loss can be described by a fractional release factor (FRF)

$$\alpha = 1 - \frac{\chi}{\chi_{[entry]}} \, , \tag{1}$$

where $\chi$ is the observed mixing ratio in the stratosphere and $\chi_{[entry]}$ the mixing ratio at the tropical tropopause where the air enters the stratosphere. Consequently, changes in the strength and pattern of the stratospheric circulation cause changes in $\alpha$





and, in general, FRF highly correlate with AoA. On the one hand, it is more complicated for chemically active species to disentangle the effects of chemistry and transport. On the other hand, atmospheric measurements may be of higher quality,

more frequent, provide denser sampling, and cover longer time periods for other than the canonical species $SF_6$ and $CO_2$, which are the ones commonly used to investigate the BDC.

As many long-lived species are only sparsely measured, stratospheric $H_2O$ is particularly appealing as a tracer for estimating long-term trends, with a suite of measurements covering the past decades. The longest continuous in situ time series of stratospheric $H_2O$ (starting in 1980) comes from frost point hygrometer observations in Boulder, Colorado located at 40.0°N,

105.2°W. In addition, stratospheric $H_2O$ observations from different satellite platforms exist since the mid-1980s, such as SAGE II (Satellite Aerosol and Gas Experiment, covering the period of 1984-2005; e.g., McCormick, 1987; Chu et al., 1993; Rind et al., 1993; Thomason et al., 1997), HALOE (Halogen Occultation Experiment, 1991-2005; e.g., Russell III et al., 1993), MIPAS Envisat (Michelson Interferometer for Passive Atmospheric Sounding, 2002-2012; e.g., von Clarmann and Stiller, 2003; Raspollini et al., 2006; Fischer et al., 2008), ACE-FTS (Atmospheric Chemistry Experiment-Fourier Transform Spec-

trometer, 2004-2012; e.g., Bernath et al., 2005; Bernath, 2017) and Aura MLS (Microwave Limb Sounder, 2004-present; e.g., Waters et al., 1999, 2004, 2006). The different satellite observations are merged into homogeneous global data sets of high value for analysing stratospheric variability and trends (e.g. Hegglin et al., 2014; Davis et al., 2017). To this end, Hegglin et al. (2014) estimated trends of AoA from a novel merged satellite $H_2O$ data record, based on the conservation property of total water in the stratosphere (mainly the sum of $H_2O$ and two times $CH_4$; for details see Sect. 2). They showed that decreasing $H_2O$

mixing ratios in the lower stratosphere (below about 10 hPa) from the mid 1980s to 2010 and increasing $H_2O$ mixing ratios above are related to an accelerating shallow branch of the BDC (decreasing AoA below about 10 hPa) and to a decelerating deep branch of the BDC (increasing AoA above). It is, however, not straightforward to accurately determine the AoA from stratospheric $H_2O$ distributions due to the complex processes involved.

We consider that at a particular time and location of the stratosphere, $H_2O$ mixing ratios are determined by the value of

the stratospheric entry mixing ratio, the propagation of this entry value into the stratosphere, and the chemical source of stratospheric $H_2O$. This assumption does not hold in the lowermost stratosphere as convection and isentropic transport can cause multiple entry mixing ratios. Also in our consideration is meant that there are no any significant source processes other than $CH_4$ oxidation (e.g., neglecting all other hydrocarbons and molecular hydrogen).

The chemical source of stratospheric $H_2O$ from the oxidation of $CH_4$ is done by $O(^1D)$, OH, and Cl radicals (e.g. Röckmann

et al., 2004). The strength of the chemical source of $H_2O$ depends on the transit time and the transit path of air since entering the stratosphere and is thus related to AoA. The full complexity of these processes is very challenging to represent in the analysis of stratospheric $H_2O$. Consequently, stratospheric $H_2O$ and $CH_4$ are used in combination for AoA estimation, and the assumptions of (i) an instantaneous propagation of stratospheric entry mixing ratios, and (ii) stationarity of the correlation between AoA and the fractional release factor of $CH_4$, are frequently assumed approximations (e.g. Schoeberl et al., 2000,

2005; Hegglin et al., 2014). Hegglin et al. (2014) in particular point out that the assumption of stationarity is only valid due to the lack of a trend in tropospheric $CH_4$ over the specific period they considered.



In this paper, we investigate the methodology to estimate AoA trends from stratospheric $H_2O$ and $CH_4$. In particular, we address in detail the impact of the two frequently employed approximations described above, of (i) instantaneous entry mixing ratio propagation, and (ii) constant FRF-AoA correlation. For this purpose, we employ a closed model environment (the "model world") in which the mean stratospheric AoA and its trend is known, which is not the case when atmospheric measurements are analysed. In this way, the effect of each approximation on the calculated AoA trend and the associated temporal development of $H_2O$ can be quantified ("proof of concept"). Our results highlight the importance of assessing the robustness of observation-based methods against uncertainties in the underlying assumptions to test their validity and general applicability.

Section 2 introduces the method, in particular describing the Chemical Lagrangian Model of the Stratosphere (CLaMS, McKenna et al., 2002a; Pommrich et al., 2014) which is employed here as the model framework to simulate stratospheric AoA. Section 3 presents the results, Sect. 4 the discussion, and Sect. 5 contains the overall conclusions.

## 2 Methods

### 2.1 The CLaMS model

This paper is based on a study performed within the "model world" using the Chemical Lagrangian Model of the Stratosphere (CLaMS McKenna et al., 2002a,b). The model set-up is described in detail elsewhere (Poshyvailo et al., 2018). Briefly, CLaMS is a modular Lagrangian chemistry transport model based on 3D-forward trajectories with parametrisation of small-scale mixing. CLaMS consists of several modules, such as Lagrangian advection (TRAJ), stratospheric mixing (MIX), sinks of $H_2O$ (CIRRUS), stratospheric chemistry, and several other modules responsible for simulation of various physical and chemical processes. The modules act successively at each time step of 24 h.

The CLaMS trajectory module TRAJ performs fully Lagrangian 3-dimensional advection of an ensemble of approximately 2 million air parcels. The position of each air parcel is defined in hybrid isentropic coordinates (Pommrich et al., 2014) and longitude-latitude space. Horizontal resolution is about 100 km, while the vertical resolution is defined via a critical aspect ratio, of about 250 (Haynes and Anglade, 1997). The CLaMS simulations cover the atmosphere from the surface to the stratopause (2500 K or $\approx$ 60 km). The advection of forward trajectories in CLaMS is calculated using 6-hourly wind fields and total diabatic heating rates from meteorological ERA-Interim reanalysis (Dee et al., 2011). Wind fields are linearly interpolated from the adjacent grid points to the locations of the air parcels. The trajectory calculation advects air parcels to new positions within one model time step.

The parametrization of small-scale mixing in the CLaMS mixing module is based on the deformation rate in the large-scale flow: air parcels may be merged, or new air parcels may be inserted at each time step, depending on the critical distances between the air parcels (McKenna et al., 2002a; Konopka et al., 2004). Note, that the mixing parametrization affects both horizontal (associated with deformation in the horizontal flow) and vertical (related to the vertical shear) diffusivity (Konopka et al., 2004, 2005).

Dehydration in CLaMS is performed with the CIRRUS module (e.g., Poshyvailo et al., 2018). The calculation includes freeze-drying in regions of cold temperatures, mainly occurring around the tropical tropopause and in the Southern polar vortex.





These cold temperatures cause formation and sedimentation of ice particles. If saturation along a CLaMS air parcel trajectory exceeds a critical saturation, then the $H_2O$ amount in excess is instantaneously transformed to the ice phase and sediments out. The parametrization of sedimentation is based on a mean ice particle radius, the characteristic sedimentation length and the corresponding fall speed. The fall distance of the ice particles is calculated from the fall speed and the computation time step. After comparison of the fallen path with a characteristic sedimentation length of the vertical grid size, a respective fraction

of ice will be removed. If the parcel is sub-saturated and ice exists, the ice will be instantaneously evaporated to maintain saturation. For further details see Poshyvailo et al. (2018). $CH_4$ oxidation is included in CLaMS as a source of $H_2O$ in the middle and upper stratosphere (for details see Pommrich et al., 2014). Note, that due to the simple parametrization of ice microphysics and the omission of a parametrization of convective processes, the simulated $H_2O$ results are meaningful only above the tropopause.

## 2.2 Age spectrum calculated with CLaMS

In general, the mixing ratio of any long-lived trace gas $\chi(r, t)$ at a specific time and specific location in the stratosphere, with assumed absence of integrated loss, can be expressed as the following integral over all the past times (Hall and Plumb, 1994; Waugh and Hall, 2002)

$$\chi(r, t) = \int\limits_{0}^{\infty} \chi(r_0, t - t') \, G(r, t \mid r_0, t - t') \, dt'. \tag{2}$$

where $t$ is the field time when the volume is sampled, $t'$ is the transit time; $G(r, t \mid r_0, t - t')$ is the boundary propagator or Green's function of the transport operator. Here, $G$ is interpreted as a transit time distribution (the "age spectrum"), and is the probability that the transit time of the air parcel travelling from the source $r_0$ to the sample point $r$ is in the range between $t'$ and $t' + dt'$. Moreover, the age spectrum fulfils the normalization condition that its integral over transit time is unity. The first moment of the age spectrum is the mean age of air (AoA, denoted by the symbol $\Gamma$ in the following). In this way, the

stratospheric tracer distribution can be described through the contributions of their tropospheric evolution and transport.

In our study, the age spectrum is computed with CLaMS driven by the ERA-Interim reanalysis, using the "Boundary Impulse (time-)Evolving" (BIER) method based on multiple tracer pulses (Ploeger and Birner, 2016). For the inert tracer $\chi$ with a pulse at the location $r_0$, the field time $t_0$, and the source time $t_0^*$, the time evolution of the source can be described with a $\delta$-distribution. Thus, Eq. 2 can be transformed to

$$\chi(r, t' + t_0^*) = G(r, t' + t_0^* \mid r_0, t_0^*), \tag{3}$$

where $t' = t_0 - t_0^*$ is a transit time, $G(r, t' + t_0^* \mid r_0, t_0^*)$ is the boundary impulse response at location $r$ to a $\delta$-boundary condition at the location $r_0$ at source time $t_0^*$. Having $N$ different tracers $\chi_i$ $(i = 1, ..., N)$ with pulses at the source location $r_0$ at times $t_{0[i]}$ provides the field time dependence of the propagator $G$. The age spectrum, may be constructed at each field time $t$ and location $r$ as $G(r, t \mid r_0, t - t'_{[i]}) = \chi_i(r, t)$. Hence, the $N$ different tracers provide $N$ pieces of information for the age spectrum at the

discrete transit times $t'_{[i]} = t - t_{0[i]}$.



Here, $N = 60$ different boundary pulse tracers were used. These pulses were released directly at the tropical tropopause between 30° S-30° N. Precisely, the source region covers the potential temperature layer from 10 K below to 10 K above the WMO (lapse rate) tropopause. The particular tracer mixing ratio is set to 1 for each pulse for a period of 30 days at the location $r_0$, and it is set to 0 in $r_0$ at other times. Pulses are launched every month. Consequently, to build the age spectrum for January

1990, the most recent tracer pulse has source times in January 1990, the second tracer pulse in December 1989, and so on. In our study, the original length of each age spectrum is 10 years (threshold transit time). The tail of the age spectrum is approximated with an exponential function when transit time exceeds 10 years (e.g., Ploeger and Birner, 2016). We used the exponential correction for the tail back to January 1979 for each age spectrum. After this correction, the age spectrum was normalized to unity.

## 2.3   Contributions to $H_2O$ changes

In this paper, we investigate the methodology to estimate AoA trends within a closed model environment of CLaMS in which the mean stratospheric $H_2O$ and $CH_4$ and their trend are known. Trends in AoA can be calculated from trends in stratospheric $H_2O$ mixing ratios by using the conservation property of total hydrogen in the stratosphere, namely that the sum of $H_2O$ and two times $CH_4$ mixing ratios is approximately constant (e.g., Le Texier et al., 1988; Dessler et al., 1994). This conservation

property implies for the $H_2O$ mixing ratio at a given location $r$ and time $t$ in the stratosphere

$$H_2O(r,\ t) = H_2O_{[entry]}(r,\ t) + 2\alpha(r,\ t)\ CH_{4[entry]}(r,\ t), \tag{4}$$

where $H_2O_{[entry]}(r,\ t)$ and $CH_{4[entry]}(r,\ t)$ are $H_2O$ and $CH_4$ mixing ratios respectively at the specific location and time in the stratosphere, being transported from their entry location at the tropical tropopause without any chemical effects.

The time series of CLaMS $H_2O$ and $CH_4$ at the entry to the stratosphere (averaged over the tropics at 30°S-30°N and in the

potential temperature layer 390-400 K, chosen just above the cold tropical tropopause region.) is shown in Fig. 1a, b. While the $CH_4$ time series shows characteristics close to clock tracers, the $H_2O$ time series is highly variable. Annual zonal mean of stratospheric $H_2O$ and $CH_4$ from CLaMS is shown in Fig. 1c, d. As already mentioned, stratospheric $H_2O$ is controlled mainly by CH4 oxidation in the middle and high stratosphere, while local freeze-drying dominates in the Antarctic polar vortex. Consequently, $CH_4$ mixing ratios generally decrease with increasing altitude, as it is gradually chemically transformed into

$H_2O$.

The fractional release factor (FRF), $\alpha$, describes the fraction of $CH_4$ which has been dissociated in the stratosphere (Solomon and Albritton, 1992), and it can be determined as

$$\alpha(r,\ t) = 1 - CH_4(r,\ t)/CH_{4[entry]}(r,\ t), \tag{5}$$

As $CH_4$ oxidation depends on the altitude in the stratosphere, the FRF is strongly affected by the depths of the BDC. Hence,

information on circulation trends (in particular on AoA) can be deduced from trends in $\alpha$ (Hegglin et al., 2014). Assuming long-term trends as small perturbations to the basic state, Eq. 4 can be rewritten as an equation for the linear trend in stratospheric





$H_2O$ over a given time period

$$\Delta H_2O(r,\,t) = \Delta H_2O_{[entry]}(r,\,t) + 2\alpha(r,\,t)\,\Delta CH_{4\,[entry]}(r,\,t) + 2CH_{4\,[entry]}(r,\,t)\,\Delta\alpha(r,\,t). \tag{6}$$

Here, $\Delta H_2O_{[entry]}(r,\,t)$ and $\Delta CH_{4\,[entry]}(r,\,t)$ are the trends in water vapour and methane transported from their stratospheric entry location at the TTL (neglecting chemistry), and $\Delta\alpha(r,\,t)$ is the trend in FRF. Note that the sum of $H_2O$ and two times $CH_4$ mixing ratios is not conserved around and below the tropopause and in the polar vortex where dehydration causes loss of $H_2O$. Hence, the presented analysis does not apply at the lowermost stratosphere, below the tropopause and in the polar vortex or near its edges.

Using the above approach, it is possible to investigate the different contributions to stratospheric $H_2O$ changes related to changes in stratospheric $H_2O$ entry mixing ratio, changes in $CH_4$ entry mixing ratio, and changes in the FRF. As the last term represents a change in stratospheric circulation (which, in turn is related to changes in the BDC), we can estimate the changes in the AoA from $\Delta\alpha$. In the next sections, we analyze in detail the AoA trend calculations based on $H_2O$ changes within the CLaMS model world.

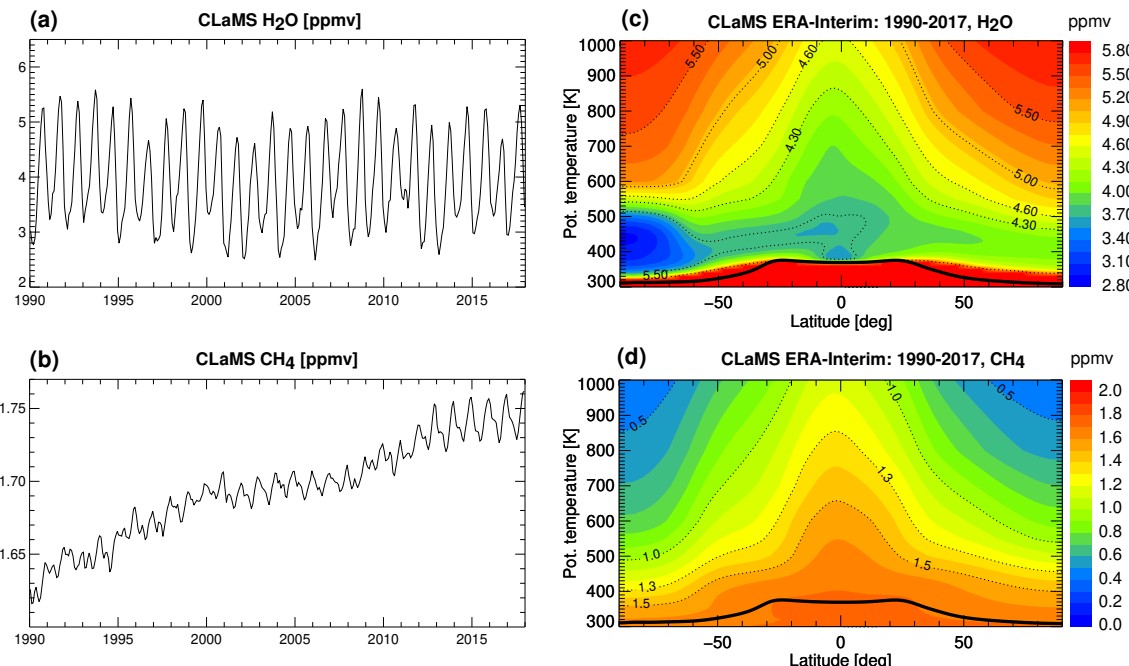

**Figure 1.** Time series of $H_2O$ (a) and $CH_4$ (b) mixing ratios (in ppmv, see y-axis) for 1990-2017 averaged over the tropics at 30°S-30°N and in the potential temperature layer 390-400 K. Annual zonal mean $H_2O$ distributions for $H_2O$ (c) and $CH_4$ (d). Data shown is from the CLaMS simulation driven by ERA-Interim reanalysis. For CLaMS, the boundary conditions at the surface are prescribed based on ground-based measurements in the lowest model level (below ≈ 4 km). For $CH_4$ from 1990 to 2011 it is taken from the zonally-symmetric NOAA/ESRL dataset (e.g., Masarie et al., 1991), for 2011-2017 it is taken from zonally-resolved AIRS data (e.g., Xiong et al., 2008, 2013).


## 2.4 Methods of assessing AoA trends from $H_2O$ changes

There are four methods of AoA trend calculation used in this study, depending on the assumed approximations of (i) an instantaneous propagation of stratospheric entry mixing ratios, and (ii) stationarity of the correlation between AoA and the

**Table 1.** Specification of the terms (first column of the table) required for AoA trends estimation from $H_2O$ changes in different methods: full reconstruction (FULL), constant correlation (C-CORR), approximation (APPROX) and improved approximation (APPROX-improved) methods.

| Term | FULL | C-CORR | APPROX | APPROX-improved |
|---|---|---|---|---|
| $\Delta H_2O_{[entry]}(r, t)$ | Trend of $H_2O$ propagated by monthly age spectrum, see Eq. 2 | Trend of $H_2O$ propagated by monthly age spectrum, see Eq. 2 | Trend from $H_2O$ time-series averaged over 390-400 K and 30° S-30° N | Trend of $H_2O$ monthly propagated by parameterised idealised age spectrum |
| $2\alpha(r, t)\,\Delta CH_{4[entry]}(r, t)$ | Trend of $CH_4$ propagated by monthly age spectrum, see Eq. 2; FRF is defined from Eq. 5 | Trend of $CH_4$ propagated by monthly age spectrum, see Eq. 2; FRF is defined from Eq. 5 | Trend from $CH_4$ time-series averaged over 390-400 K and 30° S-30° N; FRF is defined from approximated Eq. 5 | Trend of $CH_4$ monthly propagated by parameterised idealised age spectrum, see Eq. 2; FRF is defined from Eq. 5 |
| $2CH_{4[entry]}(r, t)\,\Delta\alpha(r, t)$ | Trend of FRF defined from Eq. 5; used the climatological mean of $CH_4$ propagated by monthly age spectrum | Trend of FRF is residual from Eq. 6; used the climatological mean of $CH_4$ propagated by monthly age spectrum and trend of CLaMS $H_2O$ | Trend of FRF is residual from Eq. 6; used $CH_4$ averaged over 390-400 K and 30° S-30° N and trend of CLaMS $H_2O$ | Trend of FRF is residual from Eq. 6; used the climatological mean of $CH_4$ monthly propagated by parameterised idealised age spectrum and trend of CLaMS $H_2O$ |
| $\Delta AoA$ | AoA trend is from recalculated AoA from FULL FRF: defined by the monthly varying correlation function $f$ between FULL FRF and CLaMS AoA, where $AoA = f(\alpha)$ | AoA trend is defined by the third-order polynomial constant in time empirical correlation function $f$, $\Delta AoA = f(\alpha + \Delta\alpha) - f(\alpha)$; note, that $\alpha$ is from APPROX | AoA trend is defined by the third-order polynomial constant in time empirical correlation function $f$, $\Delta AoA = f(\alpha + \Delta\alpha) - f(\alpha)$, used $\alpha, \Delta\alpha$ from this method | AoA trend is defined by the d-order polynomial constant in time empirical correlation function $f$, $\Delta AoA = f(\alpha + \Delta\alpha) - f(\alpha)$; note, that $\alpha$ is from APPROX |





FRF of $CH_4$. The summary of the different terms needed for AoA estimation with respect to the used methods is shown in Table 1.

The full reconstruction method (FULL) includes the most detailed representation of the true atmospheric processes. In the FULL method, $H_2O$ and $CH_4$ entry mixing ratios are propagated through the convolution of the TTL mixing ratios with the modelled age spectrum. Also, the monthly varying FRF-AoA correlations are used for translating FRF into AoA changes. None of the approximations (i) or (ii) is used in the FULL method. Note, that for estimating AoA trend with the FULL method, the propagated entry $H_2O$ mixing ratios are not used; the AoA trend is deduced from the FRF, which requires only propagated

entry $CH_4$ mixing ratios.

The constant correlation method (C-CORR) includes the propagation of entry $H_2O$ and $CH_4$ mixing ratios by the modelled age spectrum (same as for FULL), but the stationary relationship between FRF and modelled AoA is used (ii approximation). The difference between C-CORR and FULL is in the used correlation between AoA-FRF, and the way of FRF calculation. For C-CORR, a constant AoA-FRF was used, whereas for FULL a monthly varying correlation was used to include effects of

the non-stationarity of the correlation. The procedure of propagating entry mixing ratios in C-CORR and FULL is exactly the same.

The method based on both approximations (i) and (ii) is named "approximation method" in the following, and is abbreviated APPROX. In fact, APPROX method is used in the literature, and we evaluate it in this paper in detail. Finally, we introduce an improvement to the approximation method: instead of using approximation (i), stratospheric entry $H_2O$ and $CH_4$ mixing ratios

are propagated with the parameterised idealised age spectrum.

## 3 Results

In the following we consider the consequences of the two major approximations (i) instantaneous propagation of stratospheric entry mixing ratios, and (ii) constant correlation (stationary relationship) between FRF and AoA. We evaluate the effects of these approximations on the AoA trends inferred from $H_2O$ changes through comparison of the "true" AoA trend (actual

modeled with CLaMS) and the AoA trends estimated with the different methods (see Sec. 2.4). First, we consider the extended 1990-2017 period and thereafter 1990-2006. The results of this work provide an estimate of the reliability of the approximation method to deduce circulation trends from observed stratospheric $H_2O$ and $CH_4$ mixing ratios.

### 3.1 Contributions to stratospheric $H_2O$ trends

As described in Sect. 2.3, AoA trends can be estimated from the separation of $H_2O$ changes into different contributions. In

a first step, we estimate each term of Eq. 6 using the full reconstruction method FULL through propagation of $H_2O$ and $CH_4$ entry mixing ratios by the modelled age spectrum (see Table 1). The $H_2O$ mixing ratio transported from the location of stratospheric entry at the tropical tropopause without chemical effects constitutes the first term of this equation. Here, we estimate this term by propagating the tropopause mixing ratios (at $r_0$) to the stratospheric sampling location $r$ and time $t$ using Eq. 2, by convoluting the mixing ratio $\chi(r_0, t - t')$ at the tropical tropopause with the transport operator's Green's function,



$G(r, t; r_0, t - t')$. The Green's function $G$, or stratospheric age spectrum, has been simulated by CLaMS and is known over the considered period (for details on age spectrum calculation see Sec. 2.2). Hence, the propagation of boundary mixing ratios to each grid point in the stratosphere provides the full reconstructed stratospheric tracer field. The analogous calculation is applied to derive $CH_4$ mixing ratios in the stratosphere, transported without including chemical effects.

In our study, the entry time series of $H_2O$ and $CH_4$ are taken from zonally averaged monthly mean data simulated with
CLaMS driven by ERA-Interim reanalysis. The location of entry to the stratosphere is approximated as the 390-400 K layer between 30°S-30°N, which is located just above the cold point tropopause to avoid complications related to $H_2O$ dehydration processes at the tropopause (see Fig. 1a, b). The propagation procedure yields zonally averaged stratospheric entry $H_2O$ and $CH_4$ distributions with a monthly resolution on the latitude-potential temperature grid.

For deducing FRF, the relation from Eq. 5 is used, where $CH_4(r, t)$ is monthly mean $CH_4$ simulated with CLaMS driven by ERA-Interim reanalysis. The $CH_{4\,[entry]}(r, t)$ is calculated as described above, through the convolution of the tropical entry
mixing ratio time series with the age spectrum. Consequently, the resulting FRF has a monthly resolution.

Thus, we can estimate each contribution to the stratospheric $H_2O$ mixing ratio. All trends are calculated through a linear
regression which minimizes the standard deviation at each latitude-potential temperature grid. Note that in this study we applied the boundary time series propagation procedure with the CLaMS age spectra, on the period 1990–2017, because of the availability of the CLaMS data and the necessary age spectrum length. The different contributions to the total stratospheric $H_2O$ change for 1990-2017 are shown in Fig. 2. Figures 2a, b represent the first two terms of Eq. 6, related to the entry $H_2O$ and $CH_4$ mixing ratio trends, respectively. Figure 2c shows the impact from circulation changes, in terms of the change in FRF.
In general, the different contributions affect the stratospheric $H_2O$ changes differently in different regions, consistent with the findings of Hegglin et al. (2014). The strongest regional pattern is apparent in the contribution related to the stratospheric circulation change (Fig. 2c).

For assessing the reliability of the method applied to estimate the different contributions, we compare the stratospheric $H_2O$ trend reconstructed as the sum of the calculated contributions (in Fig. 2) with the actual trend of CLaMS simulated $H_2O$

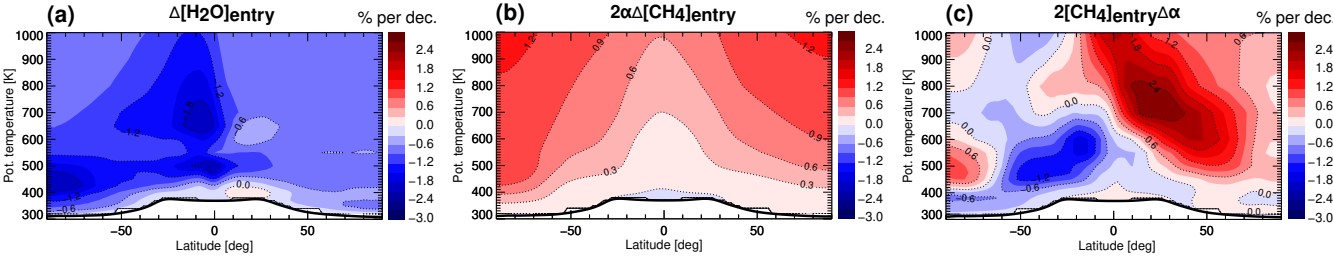

**Figure 2.** Contributions to stratospheric $H_2O$ trends for 1990-2017 from (a) stratospheric $H_2O$ entry mixing ratio changes, (b) stratospheric $CH_4$ entry mixing ratio changes, and (c) circulation changes. Stratospheric entry $H_2O$ and $CH_4$ are derived through propagation of their stratospheric entry mixing ratios by convolution with the CLaMS modelled age spectrum. The data is from CLaMS simulations driven by ERA-Interim reanalysis, and is presented in percentage per decade, with relation to the climatological 1990-2017 stratospheric $H_2O$ mixing ratios. The black line is the (llapse rate) tropopause calculated from ERA-Interim using the WMO definition (WMO, 1957).





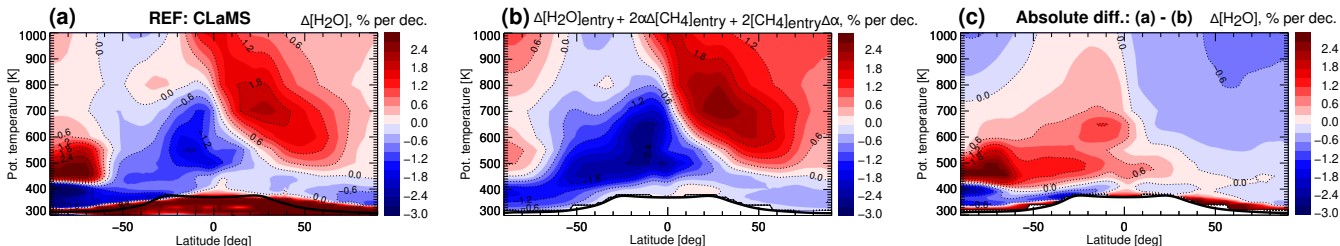

**Figure 3.** $H_2O$ trends during 1990-2017 period and their reconstruction, shown in percentage per decade with respect to the 1990-2017 climatology. Sub-figure (a) represents the reference (REF) "true" stratospheric $H_2O$ trend calculated from the CLaMS simulation driven by ERA-Interim reanalysis. (b) shows the reconstruction as sum of the different contributions (see Fig. 2), including propagated stratospheric entry $H_2O$ and $CH_4$ by the age spectrum and the change in FRF. (c) shows the absolute difference between (a) and (b). The black line is the tropopause calculated from ERA-Interim. Note, that due to the simple parametrization of ice microphysics and the omission of a parametrization of convective processes in CLaMS, the simulated $H_2O$ results (see (a)) are meaningful only above the tropopause.

(termed reference REF). Figure 3a shows the "true" $H_2O$ trend from the CLaMS simulation, while Fig. 3b shows the sum of the three terms from Fig. 2. Figure 3c shows the absolute difference between the true and reconstructed trends, indicating clear quantitative differences. Particularly large differences occur in the Antarctic region. This is expected due to the strong local dehydration occurring in that region, and the related failure of the total hydrogen conservation. Hence, the results of the reconstruction method should be interpreted with caution in the Southern polar region. Also, the disagreement between Fig. 3a

and Fig. 3b is partly related to inaccuracies in the modelled age spectrum (monthly pulsing, limited spectrum length), and to inaccuracies in the boundary time-series (averaging in the layer of 390-400 K potential temperature and 30° S-30° N). Note, that through the convolution of the age spectrum and the stratospheric entry time series it is possible to reconstruct mixing ratios only above the tropopause (or the level of the boundary time-series, if chosen differently). Outside of the Southern high-latitude regions, the overall differences shown in Fig. 3c are small, and the propagation procedure provides a good estimate of

stratospheric $H_2O$ change and its contributions, at least regarding the large-scale patterns. Note that neither stratospheric entry $H_2O$ (Fig. 2a) nor $CH_4$ changes (Fig. 2b) explain the pattern of the "true" $H_2O$ trend (Fig. 3a). Instead, the circulation change term (Fig. 2c) includes the regional characteristics of the actual stratospheric $H_2O$ trend.

### 3.2 AoA trends estimation using monthly AoA-FRF correlation

Accurate estimation of AoA from observed trace gas distributions is a complicated task. Even though, it is desirable to have

a complete age spectrum for AoA calculations, it is very difficult to obtain it from measurements. Consequently, different approximations are often applied when deriving mean AoA from trace gases observation with non-linear increase, as well as assumptions are necessary about the age spectrum and its shape (e.g., Schoeberl et al., 2000, 2005; Ehhalt et al., 2007; Hegglin et al., 2014).

The changes in FRF can be translated into changes in AoA using the correlation between estimated FRF and aforehand

known AoA, following the procedure described by Hegglin et al. (2014). In the full reconstruction method, FULL (for details





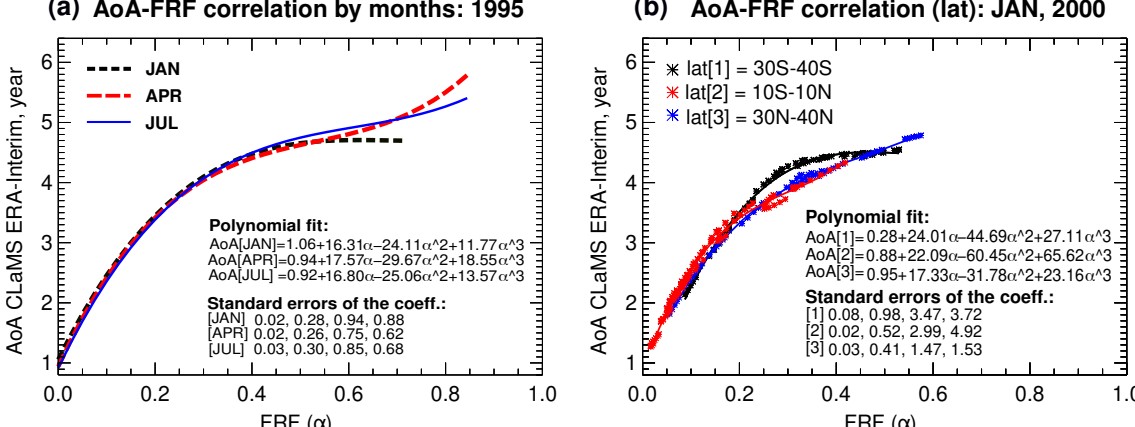

**Figure 4.** An example of relationship between estimated FRF and CLaMS AoA. On sub-figure (a) are shown correlation functions for some months of 1995: January (black dashed line), April (red dashed line) and July (blue solid line). FRF and CLaMS AoA are taken at the whole range of latitudes (90° S-90° N) for each correlation function in (a). On sub-figure (b) is shown relationship between estimated FRF and CLaMS modelled AoA for January 2000 for three intervals of latitudes: between 30° S and 40° S (black line and black star-dots), 10° S-10° N (red line and red star-dots), 30° N-40° N (blue line and blue star-dots). The estimated FRF and CLaMS AoA driven by ERA-Interim are zonally averaged data taken between 450 K and 1000 K (where FRF has positive values due to the method) in both (a) and (b). The AoA-FRF relationships are shown by monthly fitting third-order polynomial function to the AoA-FRF distribution; in (a) the actual AoA-FRF distribution is not shown to avoid an overcrowded plot.

see Table 1), a monthly varying correlation between estimated FRF and CLaMS AoA is used, where AoA $= f(\alpha)$, and $f$ is an empirically determined correlation function. In an example shown in Fig. 4 for January, April and July of 1995, the AoA-FRF correlation functions are unique for each month, because the differences in magnitude of the coefficients are greater than the standard error's range. Moreover, monthly AoA-FRF correlation functions have a very small difference for relatively young air
(< 4 years) and low FRF (< 0.4), but there are visible differences towards older AoA.

It is worth mentioning that the monthly AoA-FRF correlation functions are still a simplification and might introduce some bias in the reconstruction. In general, an accurate AoA-FRF correlation function depends not only on the considered time but also on longitude, latitude and altitude. As an example, the relationship between AoA and FRF with regard to different latitude
ranges (30° S-40° S, 10° S-10° N, 30° N-40° N) is shown in Fig. 4b for January, 2000. The AoA-FRF correlation functions are unique for each latitude range, because the differences in the magnitude of coefficients are out of the standard error's range. And, for instance, at the same FRF level of 0.3, the air at the Northern tropics (30° N-40° N) is younger than at the Southern tropics (30° S-40° S) by almost half a year.

Thus, using the monthly varying AoA-FRF correlation function (as shown on the example in Fig. 4a), FRF can be translated
into AoA. The AoA trend is calculated from the resulting AoA applying a linear fit (minimizing the standard deviation) at each latitude-potential temperature grid point. The resulting AoA trend calculated in FULL method is shown in Fig. 5b, whereas the reference "true" AoA trend calculated directly from CLaMS simulated AoA is shown in Fig. 5a. The estimated AoA trend





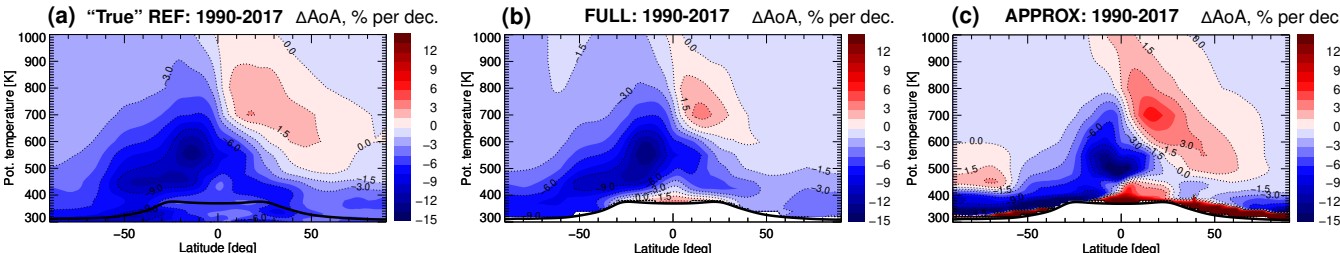

**Figure 5.** Decadal changes of AoA over 1990-2017: (a) "true" AoA changes from the CLaMS simulation driven by ERA-Interim reanalysis, (b) using $CH_4$ propagation by CLaMS age spectrum and monthly varying AoA-FRF correlation (FULL; note, that stratospheric entry $H_2O$ is not used for calculation AoA in this method), (c) AoA changes estimated with the approximation method. The changes are presented in percentage per decade relatively to the AoA 1990-2017 climatologies (regarding the used method of AoA calculation). The black line is the tropopause calculated from ERA-Interim. The white region below tropopause denotes the region, where the approximation method can not be applied.

with FULL (using $CH_4$ propagated by modelled age spectrum, and monthly varying correlation between estimated FRF and CLaMS AoA) is qualitatively and quantitatively highly reliable, when comparing to the reference CLaMS AoA trend. Note,

that for estimating AoA trends with the FULL method, the propagated entry $H_2O$ mixing ratios are not used; the AoA trend is deduced from the FRF, which requires only propagated entry $CH_4$ mixing ratios. Visible differences between Fig. 4a and Fig. 4b are related to approximations in the AoA-FRF correlation: monthly AoA-FRF correlation functions are still a simplification and might introduce some bias in the reconstruction. Also, averaged $CH_4$ boundary mixing ratios are used for reconstruction (between 390-400 K potential temperature, from 30° S to 30° N), which could induce biases as well.

## 3.3  AoA trends estimation in the approximation method (APPROX)

In the following, we further investigate the method of AoA trend estimation from combination of $H_2O$ and $CH_4$ changes, applying the two major approximations introduced above: (i) instantaneous propagation of stratospheric entry mixing ratios, and (ii) constant correlation (stationary relationship) between FRF and modelled AoA. The method based on these approximations will be termed "approximation method" (APPROX) in the following (see Table 1).

In the APPROX method, each term of Eq. 6 is approximated. The actual trend of stratospheric $H_2O$ (left side of the equation)
is assumed to be known aforehand. In our case it is the CLaMS simulated $H_2O$ change over the considered period, if the method is applied to observations it would be the observed $H_2O$ change. The first and second term of Eq. 6 refer to the changes in the stratospheric entry $H_2O$ and $CH_4$ mixing ratios, respectively. These values are obtained as a linear trend of the model entry mixing ratio time series for the considered period. The time series are averaged over the 390-400 K potential temperature layer

(approximately 80 hPa) and 30° S-30° N. Note again, that the region for averaging was chosen just above the cold tropical tropopause, important for the $H_2O$ entry value estimation (see Sect. 3.1). The FRF required for the second term is derived from Eq. 5. Therefore, following Hegglin et al. (2014) we use zonal mean simulated stratospheric $CH_4(r, t)$ mixing ratios averaged over 2005-2006, and calculate $CH_{4[entry]}$ as a mean mixing ratio between 390-400 K potential temperatures and 30° S-30° N





over 2002-2006. The impact of the circulation changes on stratospheric $H_2O$ changes is described by the change in FRF (third term in Eq. 6). For APPROX, this term is calculated as a residual between the actual CLaMS $H_2O$ trend over the considered period (left side of Eq. 6) and the other two terms of Eq. 6. Dividing the residual by $2CH_{4[entry]}$ yields the FRF changes, denoted as $\Delta\alpha$.

It is well known that FRF correlates with the AoA. Consequently, the changes in FRF can be translated into an AoA trend.

In order to estimate the AoA trend induced by the changes in stratospheric $H_2O$, we define the relationship between FRF from APPROX and modelled AoA (taken as 2005-2006 climatology from CLaMS simulation). The correlation function is derived by fitting a third-order polynomial, as suggested by Hegglin et al. (2014). In our study, the empirical relationship between APPROX FRF and CLaMS AoA is described by the function $f(\alpha) = 0.85 + 16.49\alpha - 25.30\alpha^2 + 13.77\alpha^3$ (see Appendix A, Fig. A1). As pointed out above, the approximation method assumes, that the AoA-FRF relationship is stable in time (stationary).

Note that by applying a constant AoA-FRF correlation function some atmospheric variability can be lost. Using the correlation function $f$ we obtain the AoA changes from previously estimated FRF ($\alpha$) and its changes ($\Delta\alpha$), as $\Delta AoA = f(\alpha + \Delta\alpha) - f(\alpha)$. A summary of all terms defined within the APPROX method is provided in Table 1.

The resulting decadal AoA change for 1990-2017 estimated with the approximation method is shown in Fig. 5c, and can be compared to the "true" AoA trend from the reference CLaMS simulation, shown in Fig. 5a. There are visible quantitative

differences between the "true" and the estimated trend of the AoA, especially in the Antarctic region. These differences are related to the dehydration processes occurring in that region. Also, the approximation method overestimates the AoA trend in the NH subtropical middle stratosphere. In the extratropical lowermost stratosphere (below about 380 K), AoA trends calculated with the approximation method are even opposite compared to the true trends, likely related to significant transport into this region across the subtropical tropopause (e.g., Hauck et al., 2019) which is not represented in the simplified reconstruction

here. But overall, both the estimated and the "true" AoA trend show good agreement: decreasing AoA in the LS, and increasing AoA in the NH middle stratosphere. Interestingly, for the 1990-2017 period AoA trends show clear differences between the NH and SH. The hemispheric differences might be related to the effect of mixing (e.g., Ploeger et al., 2015) and shifting stratospheric circulation patterns (Stiller et al., 2017), and were also found in long-term AoA trend derived from the observed stratospheric $CH_4$ (Remsberg, 2015). Overall, the approximation method provides a good estimate of the AoA trend for 1990-

2017, corroborating the validity of the applicability of this method to $H_2O$ observations over similar time periods (e.g., Hegglin et al., 2014).

To assess the general applicability of the approximation method, we consider another period of 1990-2006. The "true" AoA trend for this period from the CLaMS simulation is presented in Fig. 6a, and the result from the approximation method is shown

in Fig. 6c. In this case, the AoA trend from the approximation method disagrees substantially when compared with the "true" trend. Differences occur even in the sign of the AoA trend. Particularly clear differences occur in the strength of the AoA trend and its detailed pattern. Thus, the accuracy of the estimated AoA changes from the approximation method largely depends on the considered period. In the following section, we further investigate the effects of the approximations (instantaneous entry mixing ratio propagation; constant AoA-FRF correlation) and discuss their impact on the quality of the estimated AoA trend.





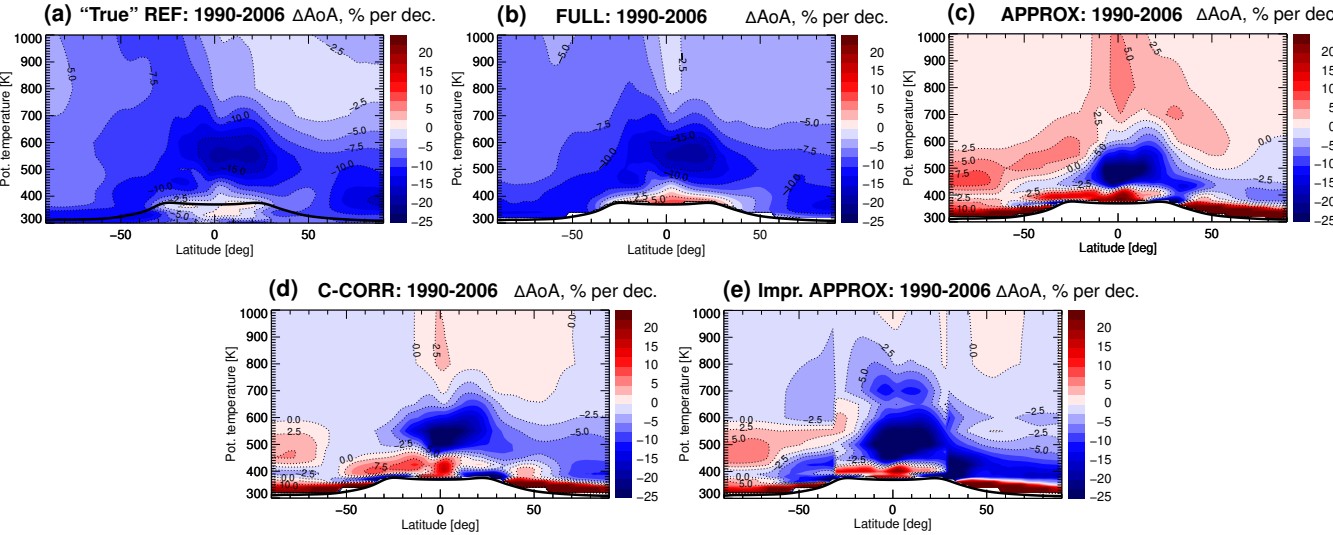

**Figure 6.** Comparison of decadal AoA trends for the 1990-2006 period, estimated using different methods: (a) "true" AoA trend from CLaMS simulation driven by ERA-Interim reanalysis, (b) using $CH_4$ propagation by CLaMS age spectrum and monthly varying AoA-FRF correlation (FULL; note, that stratospheric entry $H_2O$ is not used for calculating AoA in this method), (c) AoA trend derived from the approximation method (APPROX, instantaneous stratospheric entry $H_2O$ and $CH_4$ propagation and a constant AoA-FRF correlation), (d) using stratospheric entry $H_2O$ and $CH_4$ propagation with the CLaMS age spectrum and a constant AoA-FRF correlation (C-CORR), (e) improved approximation method (stratospheric entry $H_2O$ and $CH_4$ propagation by the parameterised age-spectrum and a constant AoA-FRF correlation. The AoA trends are presented in percentage per decade with respect to the 1990-2006 AoA climatologies from the respective method. The black line is the climatological 1990-2006 tropopause calculated from ERA-Interim. Below the tropopause, the white region indicates the areas, where the propagation procedure can not be applied.

## 3.4 Effect of the approximations: entry mixing ratio propagation and constant AoA-FRF correlation

Firstly, we evaluate the effect of approximation (i) of the instantaneous entry mixing ratio propagation. For this purpose, we perform an additional sensitivity study, where entry $H_2O$ and $CH_4$ mixing ratios are propagated through the convolution of the CLaMS TTL mixing ratios with the modelled age spectrum, but with keeping the stationary relationship between FRF and modelled AoA. This method is termed constant correlation method C-CORR in the following (see Table 1). Note that in both methods, C-CORR and APPROX, the same FRF distribution ($\alpha$) was used, but the changes in circulation ($\Delta\alpha$) are different depending on whether the stratospheric entry $H_2O$ and $CH_4$ were propagated by the age spectrum or not.

The estimated AoA trends from C-CORR for 1990-2006 are shown in Fig. 6d. The approximation of the instantaneous entry propagation largely affects the AoA trend, as evident from comparison of the resulting AoA trends from C-CORR and APPROX, as well as the CLaMS reference AoA trend (Fig. 6). Including the entry $H_2O$ and $CH_4$ propagation by the age spectrum in the method clearly improves the estimated AoA trend. When comparing C-CORR to APPROX the general





trend patterns stay similar, but improvements are visible in the extratropical lower stratosphere and above about 600 K (from comparison of C-CORR results with the true trend, see Fig. 6 d and a).

For evaluating the effect of the approximation (ii) of a constant correlation between FRF and AoA, we compare the resulting
AoA trends from C-CORR and FULL methods. As a reminder, in FULL, the entry mixing ratios are propagated through the convolution of the TTL mixing ratios with the modelled age spectrum, and a monthly varying FRF-AoA correlation was used. The difference between the results from C-CORR and FULL methods stems from the used correlation between AoA-FRF, and the explicit FRF calculation. Note that the propagated entry $H_2O$ mixing ratios are not used for estimating the AoA trend in FULL method; the AoA trend in FULL is deduced from the FRF, which requires only propagated entry $CH_4$ mixing ratios.

Keeping the entry $CH_4$ propagation by using the age spectrum and, at the same time, including monthly varying AoA-FRF correlation in FULL (Fig. 6b) results in the best estimate of the AoA trend, when compared to the "true" CLaMS AoA trend in Fig. 6a. Hence, the monthly varying AoA-FRF correlation improves the accuracy of the estimated AoA trend both qualitatively and quantitatively. As was discussed in the paper (see Sect. 2.3), stratospheric $H_2O$ is a highly variable tracer, and can lead to difficulties in estimating AoA trends. Hence, the good performance of the full reconstruction method FULL providing a
reliable trend estimate is likely related to the fact that stratospheric entry $H_2O$ mixing ratios are not influencing the calculation.

For a more precise assessment of the effects of the two approximations, the differences among AoA trends estimated with different methods (APPROX, C-CORR, FULL) and for two different periods, 1990-2017 and 1990-2006, are analysed (see Fig. 7). The difference between AoA trends from APPROX and C-CORR gives an estimate of the effect of the approximation

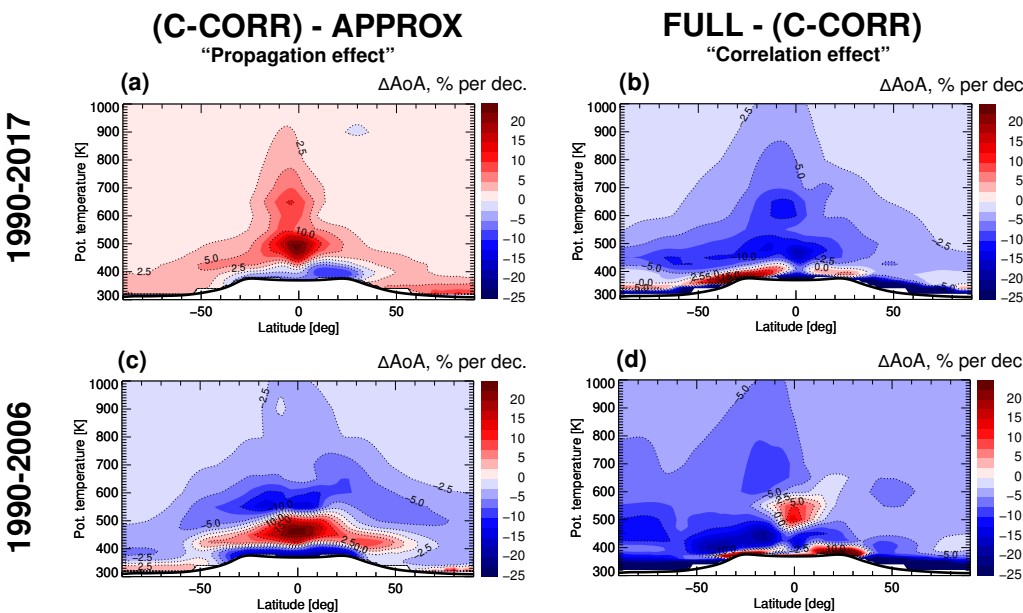

**Figure 7.** Differences in AoA trends estimated with three methods (APPROX, C-CORR, and FULL) for the periods of 1990-2017 (a, b) and 1990-2006 (c, d). The black line is the climatological tropopause calculated from ERA-Interim for the considered period. The white region below the tropopause indicates the areas where the propagation procedure can not be applied.





(i) assuming instantaneous entry mixing ratio propagation, whereas the difference between C-CORR and FULL gives an
estimate of the effect of the approximation (ii) using a constant FRF–AoA correlation.

Figure 7a, b shows differences in AoA trends for 1990-2017 between C-CORR and APPROX. The differences are less than 5% per decade above 600 K. Below 600 K, the differences in AoA trends are higher, with the maximum at approximately 480 K. Interestingly, for 1990-2017, the effects of the first and second approximations are opposite in sign. Consequently, the effects from both approximations cancel out to some extent, such that the APPROX method yields results remarkably close
to the true AoA trend (see Fig. 5). However, in general such cancellation can not be expected and the approximation method APPROX can produce misleading results.

Figure 7c, d shows the difference in AoA trends for 1990-2006. The difference above 600 K is around 5% per decade or less. A more complex structure is found below 600 K. The instantaneous entry mixing ratio propagation causes an error in the estimated stratospheric entry $H_2O$ and $CH_4$ contributions (see the discussion on contributions to stratospheric $H_2O$ in Sect. 4).
This, in turn, causes an error in the derived circulation impact, which translates through the constant AoA-FRF relationship into an error in the estimated AoA trend.

The above analysis shows that the effects of both approximations (instantaneous propagation of stratospheric $H_2O$ and $CH_4$ entry mixing ratios, and a constant correlation between FRF and AoA) on the estimated AoA trend are comparable in magnitude, however they depend on the exact period considered. Interestingly, the effects of the approximations can be opposite
in sign cancelling out each other to some extent. Consequently, the approximation method can lead to a reliable estimation of AoA trends for certain periods, but this should not always be expected. A further improvement of the approximation method is proposed in the following Sect. 3.5.

### 3.5   Improved trend estimate using parameterised age spectra

As was mentioned above, for estimating AoA trends from $H_2O$ observations, the approximations (i) instantaneous entry mixing
ratio propagation, and (ii) constant FRF-AoA correlation are necessary as the full AoA spectrum is not known in this case. However, the reliability of AoA trends estimated from $H_2O$ observations can be achieved if the used approximations are improved. As a simple and practical improvement, we propose to use an analytical, parameterised age spectrum for propagating stratospheric entry $H_2O$ and $CH_4$ mixing ratios. Note that for a further improvement of AoA trend estimates, a non-stationary FRF-AoA relationship in principle would be needed as well. But due to the sparseness of available stratospheric $CH_4$ measure-
ments deducing such a relationship from observations is challenging, and we refrain from including it in the methodological improvement. In the following, we discuss the results of an additional sensitivity study with stratospheric entry $H_2O$ and $CH_4$ mixing ratios propagated by the parameterised idealised age spectrum, and using a constant FRF-AoA correlation. This method is hereinafter referred to as the "improved approximation method" (see Table 1 for details).

In the improved approximation method we use an inverse Gaussian distribution (e.g., Newman et al., 2007; Bönisch et al.,
2009; Hauck et al., 2019) as a parameterised age spectrum (complete transit time distribution)

$$G(t, \Gamma) = \sqrt{\frac{\Gamma^3}{4\pi\Delta^2 t^3}} \cdot \exp\left(\frac{\Gamma \cdot (t - \Gamma)^2}{4\Delta^2 t}\right), \tag{7}$$





where $\Gamma$ is the mean AoA and $\Delta$ is the width of the age spectrum. Here, we parameterise AoA in different zones or "regions" depending on the considered latitude, longitude and height. The finer the separation into different regions, the less pronounced the discontinuities at the edges of the regions are. Here, we divide the stratosphere into seven regions, prescribing one value of

AoA for each region (see Appendix B). We apply the empirical relation between the spectrum width and mean AoA proposed by Hall and Plumb (1994) and used in several other studies (e.g., Engel et al., 2002; Bönisch et al., 2009)

$$\frac{\Delta^2}{\Gamma} = C, \tag{8}$$

with the constant $C = 0.7$ years, although we note that recent work suggests a larger value (2.0 years) for the lower stratosphere (Hauck et al., 2019).

The resulting AoA trend estimated with the improved approximation method for 1990-2006 is shown in Fig. 6e. There is a clear improvement in the AoA trend estimation when compared to the pure approximation method APPROX (Fig. 6c). Note, that in both methods the same FRF distribution ($\alpha$) is used, but the changes in circulation ($\Delta\alpha$) are different depending on the propagation of stratospheric entry $H_2O$ and $CH_4$. The discrepancies in estimated AoA trends between APPROX and the improved approximation method are caused by the circulation component (third term in Eq. 6). In turn, the differences in the

circulation components are caused by the discrepancies in the stratospheric entry $H_2O$ and $CH_4$ mixing ratio contributions, with the major impact of the stratospheric entry $H_2O$ trend (see Discussion in Sect. 4). The propagation of stratospheric entry $H_2O$ and $CH_4$ by the proposed parameterised idealised age spectrum results in AoA trends close to those from the C-CORR method, which is the best estimate possible for the improved approximation method due to the usage of constant FRF-AoA correlation. Thus, the improved approximation method results in a higher reliability of the estimated AoA trend, than method

the approximation method APPROX.

    Hence, we encourage usage of the improved approximation method when estimating AoA from $H_2O$ observational data. Stratospheric entry mixing ratio time series for $H_2O$ and $CH_4$ (e.g., in the TTL) can be deduced from satellite measurements, such as ACE-FTS, HALOE, MIPAS, or SCIAMACHY (e.g., Bernath, 2017; Russell III et al., 1993; Nassar et al., 2005; Raspollini et al., 2006; Scherer et al., 2008; Müller et al., 2016; Lossow et al., 2017; Noël et al., 2018). Due to the limited

stratospheric $CH_4$ observations, the stratospheric entry $CH_4$ mixing ratio contribution can be kept as in APPROX (instantaneous propagation), because this term has only little effect on the resulting AoA trend (see Discussion in Sect. 4). But the correct representation of the stratospheric entry $H_2O$ mixing ratio is very important for a reliable estimation of AoA trends. Our study shows that usage of the parameterized idealized age spectrum clearly improves the representation of the stratospheric entry $H_2O$ term, and hence the final AoA trend estimate.

**4 Discussion**

The contributions of stratospheric entry $H_2O$ mixing ratio trends (corresponding to the first term on the right-hand side of Eq. 6) calculated with the approximation method (APPROX), improved approximation and full reconstruction (FULL) methods as well as the "true" stratospheric $H_2O$ trend from CLaMS simulation are shown in Fig. 8. Note that the stratospheric entry $H_2O$

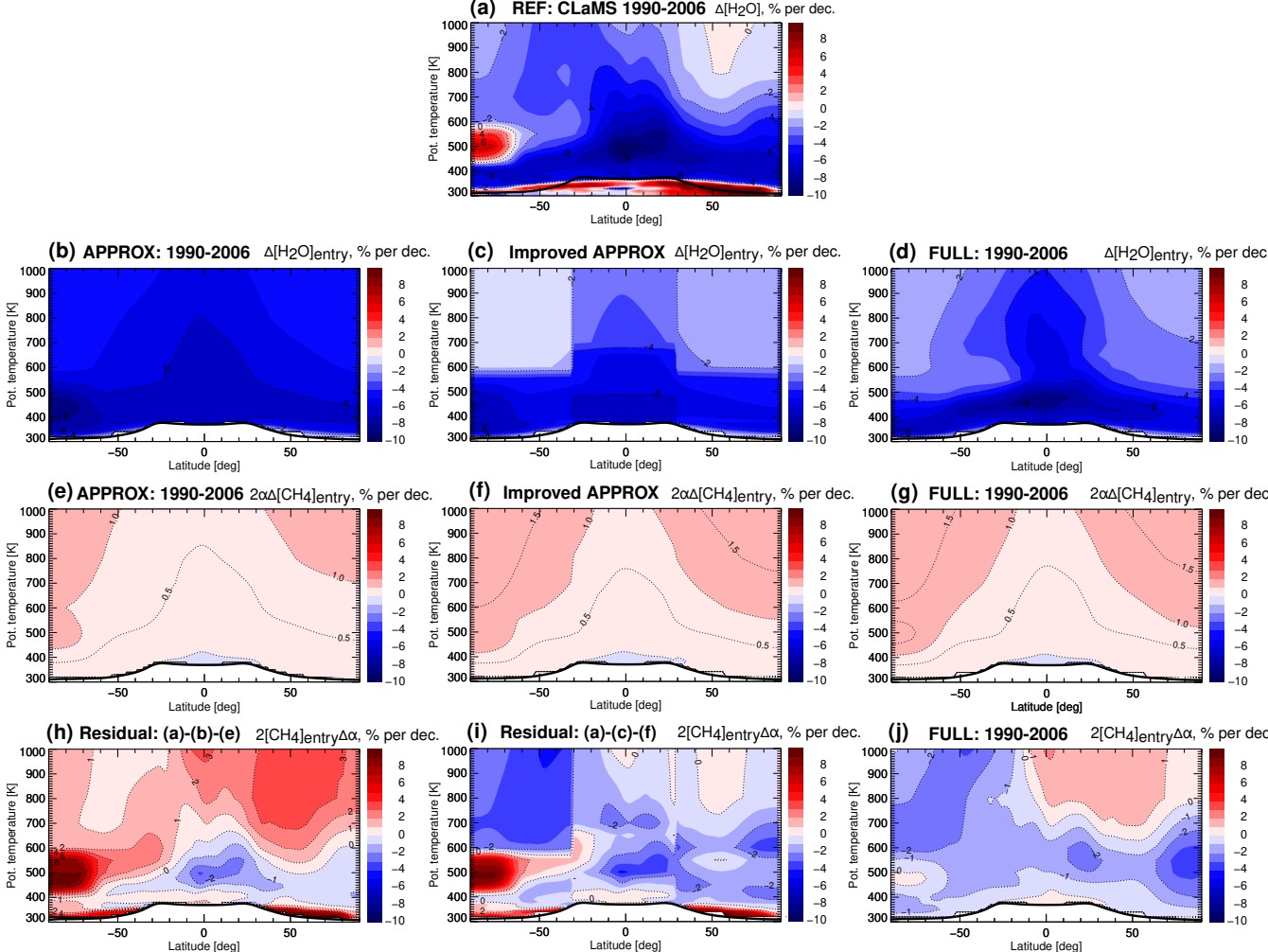

**Figure 8.** The "true" stratospheric $H_2O$ 1990-2006 trend calculated from CLaMS simulations driven by ERA-Interim reanalysis is shown in (a). Further plots represent contributions to $H_2O$ trend calculated through different methods: from the approximation method (APPROX) is shown in (b, e, h), the improved approximation method (c, f, i), and the full reconstruction (FULL) method (d, g, j). The contributions from circulation changes are calculated as the residual in APPROX and improved approximation methods (h, i), and as a linear trend in FULL (j). Note, that all sub-figures are presented in percentage per decade, with relation to the climatological mean 1990-2006 CLaMS stratospheric $H_2O$ mixing ratio. The black line is the tropopause calculated from ERA-Interim.

mixing ratio propagated by the CLaMS age spectrum (Fig. 8d) is shown here only for comparison and is not used for AoA
trend calculations in FULL method.

The stratospheric entry $H_2O$ mixing ratio trends from the approximate methods are evaluated against the respective trend from the full reconstruction FULL in Fig.8 b-d. Comparison of Fig. 8b and Fig. 8d shows that APPROX overestimates the stratospheric entry $H_2O$ mixing ratio trend by approximately 3% per decade, especially in the middle and upper stratosphere.





The propagation of stratospheric entry $H_2O$ by the idealised age spectrum improves the representation of stratospheric entry
$H_2O$ trend (see Fig. 8c, d), and thus leads to a more reliable AoA trend estimation.

The contributions from stratospheric entry $CH_4$ mixing ratio trends are presented in Fig. 8 e-g. The differences between the
various stratospheric entry $CH_4$ trends are below 0.5% per decade. Although, the pattern of the stratospheric entry $CH_4$ mixing
ratio trends is slightly improved when including propagation with the idealised age spectrum (Fig. 8f, g). But overall, this term

has only a weak effect on the circulation contribution and the resulting AoA trend.

The circulation contributions in the APPROX and improved approximation methods are calculated as the residual between
CLaMS $H_2O$ trend (Fig. 8a) and the other two components. The resulting circulation terms are shown in Fig. 8h, i. The circu-
lation change from FULL (using propagated entry $CH_4$ mixing ratio for FRF) is presented in Fig. 8j. As the full reconstruction
FULL is the most exact method, the circulation terms from the approximate methods are evaluated against it. The FRF trend

($\Delta\alpha$) in FULL is calculated as a linear trend at each latitude-potential temperature grid.

Figure 8 h and j shows that the error in the estimated circulation contribution in APPROX is large, and the sign of the
circulation contributions is even opposite in particular regions. Propagating stratospheric entry $H_2O$ and $CH_4$ by the idealised
age spectrum (Fig. 8i) improves the representation of circulation change significantly. From the comparison of Fig. 8i, j, we
conclude that calculating the circulation impact as a residual yields a reliable representation of the circulation contribution

when an idealised parameterised age spectrum is used. Large discrepancies still occur in the Antarctic region where the re-
construction method is expected to fail because of local dehydration processes. The polar dehydration in the Antarctic region
has a substantial drying effect, reaching 1 ppmv and even more below 600 K potential temperature in the Southern hemisphere
(Poshyvailo, 2020). This dehydration effect induces discrepancies between the APPROX, improved approximation and FULL
methods. Thus, the AoA trend estimated from the circulation contribution as a residual (APPROX, improved approximation

method, C-CORR), is not reliable in the Southern polar region. Furthermore, the circulation contribution calculated as a resid-
ual depends also on the accuracy of the used stratospheric $H_2O$ trend (here, the stratospheric $H_2O$ trend is calculated from
CLaMS simulation driven by the ERA-Interim reanalysis, Fig. 8a).

In summary, the differences in the circulation components in the APPROX and improved approximation methods are caused
by the discrepancies in the stratospheric entry $H_2O$ and $CH_4$ mixing ratio contributions, with the major impact from the

stratospheric entry $H_2O$ trend. The differences in the second term – contributions from stratospheric entry $CH_4$ mixing ratio
trends – are small (Fig. 8e, f, g). Consequently, the correct representation of the stratospheric entry $H_2O$ mixing ratio trend is
crucial for AoA trend estimation.

On the one hand, we showed that biases in AoA trends estimated from stratospheric $H_2O$ can be large, questioning the
usefulness of this approach. On the other hand, also age estimates based on other trace gas species, like $SF_6$ or $CO_2$, show

substantial uncertainties, resulting from the method to calculate mean age and from the behaviour of the employed trace gas
(e.g., Engel et al., 2009; Fritsch et al., 2020). An advantage of using $H_2O$ is the existence of several long and homogenized
records of satellite measurements of comparatively high quality. Furthermore, the significant bias reduction in estimated AoA
trends, related to the relatively simple methodological improvement by using an idealized age spectrum for entry mixing ratio
propagation, seems very promising. We strongly recommend that such a refinement of the method should be used for future



studies. Also, further improvements could be realized by including a chemistry-dependent propagator instead of the idealized age spectrum (see Ostermöller et al., 2017), or by using an inversion algorithm for fitting the parameterized age spectrum (e.g., Hauck et al., 2019).

## 5   Conclusions

We investigated the effects of commonly used approximations to estimate long-term Brewer-Dobson circulation changes from
stratospheric $H_2O$ by deducing mean age of air. For this reason, we designed a study within the "model world" of the Chemical Lagrangian Model of the Stratosphere (CLaMS), considering as case studies the 1990-2006 and 1990-2017 periods. Two approximations were considered: (i) instantaneous stratospheric entry mixing ratio propagation, and (ii) stationary AoA-FRF correlation. Suitable sensitivity studies were carried out to investigate the effects of these approximations, namely the approximation method (APPROX) which includes both approximations (i and ii), the constant-correlation method C-CORR which
includes only the constant correlation approximation (ii), and a full reconstruction (FULL) which includes propagation of entry mixing ratios by the modelled age spectrum and the non-stationary AoA-FRF correlation. The sensitivity experiments were analyzed through comparison with the reference CLaMS AoA trend, representing the "true" change in the model world.

The results show that both approximations (i) and (ii) have an important impact on the calculated AoA trend. These approximations strongly affect the estimated AoA trend, causing discrepancies of up to 5 % per decade above 600 K, and higher
discrepancies, 10 % per decade or even more, below 600 K. Thus, the APPROX method leads to noticeable differences in the derived mean AoA trend when compared with the "true" AoA trend from the CLaMS reference simulation. These differences are dependent on the considered period. For some periods, significant differences result in the strength and the detailed structure of estimated AoA trends. In specific regions of the stratosphere, the derived mean AoA trends from APPROX are even different in sign, resulting in a strong bias of estimated BDC changes. Depending on the period, the effects from both approximations
can also be opposite, and may even cancel out to some extent, producing an estimation remarkably close to the "true" AoA trend.

In order to increase the reliability of the derived AoA trend, we propose an improvement to the APPROX method: instead of instantaneous stratospheric entry mixing ratio propagation, the stratospheric entry mixing ratios are propagated with an idealised parameterised age spectrum. Such propagation of stratospheric entry $H_2O$ and $CH_4$ by the idealised age spectrum results
in a higher reliability of the estimated AoA trend compared to the APPROX method, especially by improving the representation of the stratospheric entry $H_2O$ contribution. Moreover, we showed that the stratospheric entry $H_2O$ contribution has a major impact on AoA trends when the circulation contribution is calculated as a residual (e.g., in the APPROX method), while the discrepancies in stratospheric entry $CH_4$ contributions do not cause large effects on the resulting AoA trend. Consequently, the correct representation of stratospheric entry $H_2O$ mixing ratios is crucial for AoA trend estimation in the approximation
methods, and the usage of, at least a parameterised and idealised, age spectrum is beneficial, particularly when estimating AoA trends from $H_2O$ and $CH_4$ observation data sets. The results from this article are of particular relevance for assessing





the uncertainty in estimates of stratospheric circulation and BDC changes from global satellite measurements of stratospheric $H_2O$.

## Appendix A: Stationary correlation function between FRF and modelled AoA

In order to estimate the AoA trend induced by the changes in stratospheric $H_2O$ on approximative methods, we define a stationary relationship between zonally averaged FRF and AoA from CLaMS simulations driven by ERA-Interim reanalysis, the averaging is done over 2005-2006 following Hegglin et al. (2014).

In the study of Hegglin et al. (2014), $CH_4$ mixing ratios averaged over 2005-2006 were used for FRF calculations. This
specific period is characterised by relatively constant tropospheric $CH_4$ values. Accordingly, the dependence of stratospheric $CH_4$ entry mixing ratios on transit time can be neglected, and $CH_4$ can be assumed constant over these years. The air needs some years to be transported through the stratosphere, and stratospheric $CH_4$ during 2005-2006 originated in the TTL during approximately 2002-2006. So, $CH_{4[entry]}$ can be calculated as a mean over the 2002-2006 period. Consequently, FRF is stable in time and does not depend on the chosen period.

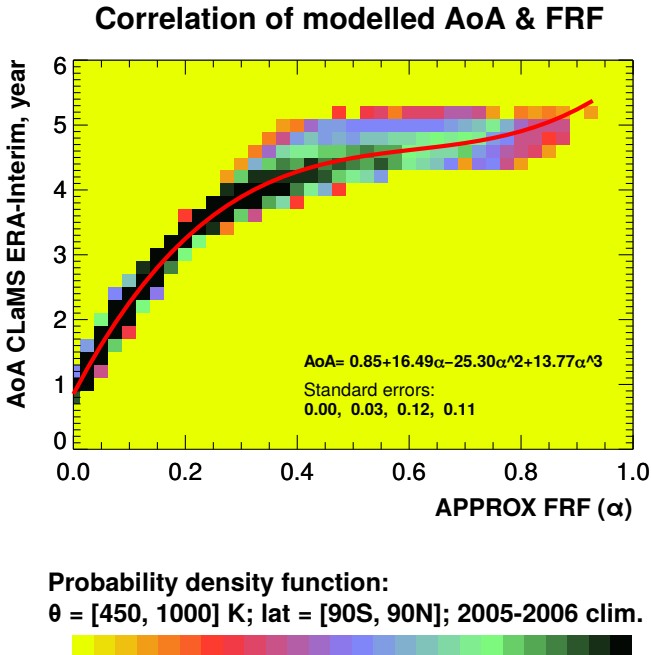

**Figure A1.** Relationship between CLaMS AoA and FRF, calculated with the approximation method. The considered correlated region is between 450-1000 K and 90° S-90° N. The colour bar represents the probability density function. The red line is the third-order polynomial fitting function. Note that due to the approximations in the method, FRF is calculated from climatological 2005-2006 stratospheric $CH_4$. Thus here, FRF is correlated with climatological AoA from the same 2005-2006 period.





The correlation function is presented in Fig. A1. It is derived by fitting a third-order polynomial, as suggested by Hegglin et al. (2014). From the Fig. A1, the empirical relationship between the CLaMS AoA and the FRF is $f(\alpha) = 0.85 + 16.49\alpha - 25.30\alpha^2 + 13.77\alpha^3$. As the approximation method assumes that the AoA-FRF relationship is stable in time, the same relationship is applied for any investigated period.

**Appendix B: Parameterization of AoA for idealised age spectrum calculations**

In the improved approximation method, we use idealised age spectrum for specified values of AoA. For this, we separated the stratosphere (up to 1000 K or approximately 37 km) in seven zones, see Fig. B1. However, further improvement by other zone divisions is, in principle, possible. A better spatial resolution of propagated $H_2O$, $CH_4$ by idealised age spectrum, and consequently estimated AoA could be gained by using more zones or even by assigning different shape for zones (e.g. triangles).

In the latitudinal directions, the middle zone between 30° S to 30° N is associated with the tropical pipe region. In the height-
direction at the units of potential temperature, the lowest AoA value of 1 year is located just above the TTL between 400-500 K (or about 15-20 km). The mean value of AoA equaled to 2.5 years, we assign to three regions: in both hemispheres between 380-600 K (approx. 12-24 km), and between 500-700 K (approx. 20-28 km) at the tropics. The averaged AoA value is 3.5 years at the tropical middle stratosphere from 700 to 1000 K (approx. 28-37 km). At the region between 600-1000 K (approx. 24-37 km) the averaged AoA value is different for both hemispheres: 4.5 years for the Southern and 4.0 years for the Northern
hemisphere. Such AoA asymmetry exists, since the deep branch of the circulation is stronger in the Northern hemisphere and thus, causing younger AoA (Butchart, 2014; Konopka et al., 2015).

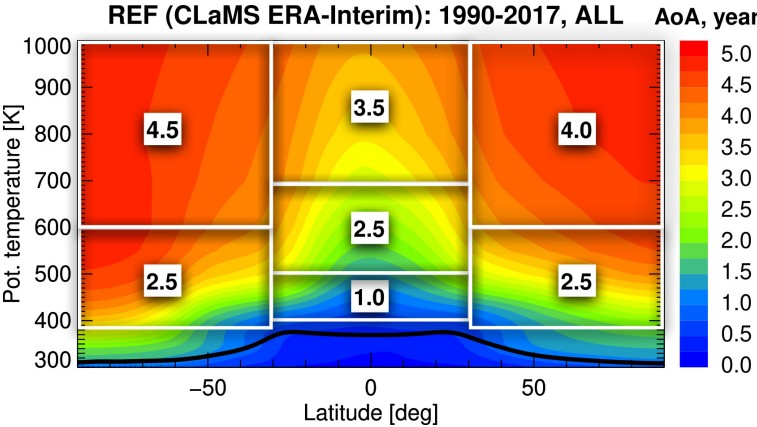

**Figure B1.** Zonal mean of AoA from CLaMS simulation driven by ERA-Interim reanalysis. Data shown are climatologies for 1990-2017 years. Regions defined with the white boxes are used for the idealized age-spectrum calculations in the APPROX-improved method. Values denoted in each box is an approximate averaged AoA value of the region bounded by white box.



*Code and data availability.* The CLaMS code used in this paper is available on theGitLab server: https://jugit.fz-juelich.de/clams/CLaMS.

*Author contributions.* The study was designed by F.P. and L.P.-S. with contributions by S.F. and R.M.. L.P.-S. and F.P. conducted the model runs and analysed the results. Partially, the work is based on the original concept from the study of M.H.. L.P.-S. wrote the paper with further inputs from F.P. and R.M. All co-authors contributed to the interpretation of the results, active discussions, and the revision of the paper.

*Competing interests.* R.M. is an editor of ACP, otherwise there are no competing interests.

*Acknowledgements.* We thank Jens-Uwe Grooß, Paul Konopka and Mengchu Tao for helpful discussions. We are also very grateful to the ECMWF for providing the reanalysis data (ERA-Interim). In addition, we gratefully acknowledge the computing time granted on the supercomputer JURECA at Jülich Supercomputing Centre (JSC) under the VSRproject ID JICG11. This work was partly funded by the German Ministry of Education and Research under grant no. 01LG1222A (ROMIC-TRIP), and partly by the Helmholtz Young Investigators Group A-SPECi ("Assessment of stratospheric processes and their effects on climate variability") under grant no. VH-NG-1128. The study conducted at Princeton University during the exchange research stay of L. Poshyvailo-Strube was partially supported by the Helmholtz Graduate School for Energy and Climate Research (HITEC) of Forschungszentrum Jülich.





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
