# Peer review of "How can Brewer-Dobson circulation trends be estimated from changes in stratospheric water vapour and methane?"

_Atmospheric Chemistry and Physics, 2021_

## Author Comment (AC1)

**Reply to reviewer #1**
**Estimating Brewer-Dobson circulation trends from changes in stratospheric water vapour and methane**

We thank the referee for the review and for the helpful and detailed comments. We give a point-by-point reply below, where the reviewer comments are repeated in black. The replies to the reviewer's comments are in blue text color. The revised text is given in italics and in quotation marks, with the positions of the corrected sentences in the revised version noted in brackets.

**General remarks**

The analysis is well done and may help with deducing trends in the stratospheric circulation from trace gas observations, which has been notoriously difficult. The topic is appropriate for ACP, the methods are clearly described and the figures are well done. I suggest the manuscript be accepted for publication with consideration of the minor comments below.

We thank the reviewer for this positive comment. In the revised version all suggested comments have been taken into account. In particular, we improved formulations and wording throughout the text as well as the captions of Fig.1,2,6 and the discussion of Fig.7. We also improved the description of the choice of zones for the parameterised age spectrum analysis (Appendix B), related to the remarks of Reviewer #1.

**Specific Comments**

1. PAGE 2
   Line 43: 'CO2 has a seasonal cycle...'

   Thank you for the remark. It is corrected at the revised version (p2, L44).
   *"...$CO_2$ has a seasonal cycle..."*

2. PAGE 3
   Line 55: awkward phrasing 'for other than', maybe 'compared to' instead?

   It is corrected at the revised version (p3, L56).
   *"...longer time periods compared to the canonical species $SF_6$ and $CO_2$..."*

   Line 77: 'Also in our consideration is meant that there are no any' is hard to understand. Perhaps rephrase to 'The only significant source of H2O considered here is CH4 oxidation (e.g. we neglect all other hydrocarbons...'

   Thank you for the remark. It is corrected at the revised version (p3, L78).
   *"The only significant source of $H_2O$ considered in this work is $CH_4$ oxidation."*

3. PAGE 6

Line 179: 'depths of the BDC' is unclear. Maybe 'vertical transport by the BDC'.

It is corrected in the revised version (p6, L182).
*"...affected by the vertical transport of the BDC..."*

4. PAGE 7

Figure 1 caption: I would suggest moving the last two sentences of the caption up into Section 2.1 since they describe important details of the CH4 boundary conditions for the model run.

Thank you for the remark. The suggested text from the caption of Fig. 1 was moved to Section 2.1 (p4, L107).
*"The boundary conditions at the surface are prescribed based on ground-based measurements in the lowest model level (below $\approx 4\,km$). $CH_4$ mixing ratios are taken from the zonally-symmetric NOAA/ESRL dataset (e.g., Masarie et al., 1991) from 1990 to 2011, and from zonally-resolved AIRS data (e.g.,Xiong et al., 2008, 2013) for 2011-2017."*

Why is the CH4 boundary condition changed from NOAA to AIRS after 2011?

Thank you for this question regarding clarity of the used $CH_4$ boundary condition.
The $CH_4$ boundary condition in CLaMS has been switched in 2011 to take advantage of the better sampling of AIRS data in comparison to NOAA, although accepting the apparent discontinuity. For the results of this paper, the discontinuity has only negligible effects, as also with this boundary condition the results of the CLaMS model are internally consistent. In particular, when using age spectra (not a lag time) in the FULL and C-CORR methods the change in the $CH_4$ boundary conditions in 2011 does not affect the results. Also for the other approximation methods $CH_4$ and FRF information is mainly used from the period 2002-2006, such that the impact of the change in $CH_4$ boundary conditions in 2011 is irrelevant.

We added the respective text to the text (p4, L110).
*"The $CH_4$ boundary condition has been switched in 2011 to take advantage of the better sampling of AIRS data in comparison to NOAA, although accepting the apparent discontinuity. Because the results of the CLaMS model are internally consistent, the discontinuity has only negligible effects."*

5. PAGE 9

Line 208: remove 'used', '...and the method of calculating FRF...'

It is corrected in the revised version (p8, L209).
*"The difference between C-CORR and FULL is in the correlation between AoA-FRF and the method of calculating FRF."*

6. PAGE 10

Figure 2 caption: 'lapse' is misspelled

Corrected in the revised version (p10, caption to Fig. 2).
*"The black line is the (lapse rate) tropopause calculated from..."*

7. PAGE 15

Figure 6 caption: The text in the parenthesis after 'FULL' and 'APPROX' isn't necessary since it's partly repetitive from the Figure 5 caption and is explained in the text.

*Thank you for this remark. The suggested text was removed from the caption of Fig. 6. (p15, caption to Fig. 6).*
*"...(b) FULL, (c) APPROX, (d) C-CORR..."*

8. PAGE 16

Line 367: '...stems from the differences in the AoA-FRF correlations used in each method...'

*Corrected in the revised version (p15, L351).*
*"The difference between the results from C-CORR and FULL methods stems from the differences in the AoA-FRF correlations used in each method, and the explicit FRF calculation."*

Line 373: '...discussed earlier in the paper...'

*It is corrected at the revised version (p16, L361).*
*"It was mentioned earlier in the paper that stratospheric $H_2O$ is ..."*

Figure 7: The constant FRF-AoA correlation approximation appears to bias the AoA trend too positive over both time periods and nearly all the stratosphere. Although you don't try to improve the APPROX method with an improved treatment of the FRF-AoA correlation it seems there might be a simple correction made for the positive trend bias. The interesting aspect is that the age trend biases are largest at the youngest ages, whereas the correlations shown in Figure 4 have no discernable differences either seasonally or latitudinally at ages younger than 3-4 years. This would be worth discussing further.

*Thank you for this comment, and we agree that this is an important point.*
*It is stated in the text already that "...the difference between the results from C-CORR and FULL methods stems from the differences in the AoA-FRF correlations used in each method, and the explicit FRF calculation..." Hence, strictly speaking, the differences shown in Fig. 7b and Fig. 7d are caused not only by the AoA-FRF correlations but also by the FRF calculation approach. Further, it is mentioned in Table 1 regarding the $2CH_{4[\text{entry}]}(r,\ t)$ $\Delta\alpha(r,\ t)$ description that this term and $\Delta\alpha(r,\ t)$ is calculated differently in the case of FULL and C-CORR (e.g. in C-CORR the FRF trend is calculated as a residual). Those differences propagate to $\Delta$AoA as well.*

*Taking all of that into account, the above mentioned differences in the young air cannot be understood in a simple way. However, the large differences in the lowermost stratosphere (directly above the tropopause) should not be over-interpreted, as in this region all methods have problems (because of the time resolution of the age-spectra). Even for the FULL method the transit time resolution of the used age spectra of 1 month causes issues in the lowermost stratosphere where transit times from the tropopause are within that range. A cautionary note is included in the revised manuscript (p16, L367).*

*"The larger differences in the lowest stratosphere directly above the tropopause should not be over-interpreted as the transit time resolution of used age spectra of 1 month is too coarse for a reliable reconstruction there."*

9. PAGE 18

   Line 413, Appendix B: I don't really understand the partitioning of the age into constant values in seven zones. Why not just use the actual age at each location? Is it too computationally expensive?

   We fully agree that using the actual age at each location would significantly improve the calculation. However, our main goal with the improved approximation method is to provide a simple and practical solution for estimating AoA trend from observations which does not require a complete AoA dataset from a model (as it might not be always available). We reformulated the statement in the revised version (p17, L397) accordingly. *"For a simple and practical method without assuming a priori knowledge of model age of air, we propose to divide the stratosphere into seven regions, prescribing one mean value of AoA for each region..."*

   Line 429: Remove second 'method' in this line.

   We decided to remove the entire sentence in the revised version.

---

## Author Comment (AC2)

**Reply to reviewer #2**
**Estimating Brewer-Dobson circulation trends from changes in stratospheric water vapour and methane**

We thank the referee for the detailed review and for the helpful comments and suggestions. We give a point-by-point reply below, where the reviewer comments are repeated in black. The replies to the reviewer's comments are in blue text color. The revised text is given in italics and in quotation marks, with the positions of the corrected sentences in the revised version noted in brackets.

**General remarks**

This is an interesting and well written paper, but overly detailed. I recommend that the authors take some time to significantly reduce the content to the salient points, briefly summarizing the experiments and driving toward the main conclusions (which are a little nebulous). Concensing and consolidating the paper would improve the focus and make it more accessible to the reader.

We thank the reviewer for the encouraging comments and the advice for improvement. The suggestions have been taken into account in the revised version. As the text had already been consolidated amongst all authors in many iterations, a further shortening indeed proved to be a challenge. However, as we agree that a better focus would make the paper more accessible to readers, we have further worked on the text throughout the entire paper, improved formulations and wording and tried to shorten and focus – taking into account the reviewers comments.

**Specific Comments to the Summary**

1. "The basic idea, as I understand it, is that the authors want to use measurements of water vapor and methane to determine trends in the BDC. ... No actual observations (except boundary forcing of methane and water) are used in this paper."

   Thank you for the comment.
   The goal of the paper is to present a proof-of-concept study within an idealized model environment. We aim to give a simple and practical advice for obtaining more reliable AoA trends estimation from the observed $H_2O$ and $CH_4$, in comparison to broadly used approaches. Therefore, we test several methods within the "model world" and state the conclusions on the possible improvements to the standard approximation. We agree that this idea was not clearly stated at the beginning of the paper. We rephrased the abstract in the revised version of the paper (p1, L8).
   *"In this work, we explore how mean age of air trends can be estimated from the combination of stratospheric $H_2O$ and $CH_4$ data, by carrying out a proof-of-concept within the model environment of the Chemical Lagrangian Model of the Stratosphere (CLaMS).*

*In particular, we assess the methodological uncertainties related to the two commonly-used approximations of (i) instantaneous stratospheric entry mixing ratio propagation, and (ii) constant correlation between mean age and the fractional release factor of $CH_4$. By carrying out different sensitivity studies with CLaMS, we test different methods of the mean age of air trend estimation, and we aim to give a simple and practical advice on the adjustment of the used approximations for obtaining more reliable mean age of air trend from the measurements of $H_2O$ and $CH_4$.*"

2. The authors assess various methods of using model water vapor and methane to determine changes in the BDC, or basically AoA trends and associated errors. I liked the evaluations they produce and an analysis of various errors (Fig. 5), but I think there is WAY too much detail, and the paper could use more of a reminder of the goals in the results section. For example, near the end perhaps you should show only 3 cases – True, Full and Improve Approx. Discussion of the other cases can be put in an Appendix since the average reader will give up while wading through this material. I think about 30% of this paper could be deleted with no loss of information content.

We agree that the paper includes many technical details. We have made many internal iterations, but since the subject is a comparison of methods where there is no generally accepted "standard method", one cannot assume the reader will be able to understand what exactly is done without detailed instructions. So, it is important to provide enough details such that the results are reproducible by others. We took into account the above suggestions in the revised version and, overall, we reduced the text of the manuscript (please, see the difference .pdf file between the submitted and updated version of the manuscript).

3. It was interesting that if you assume a simple age spectrum (Eq. 7) rather than try and reconstruct it, the methodology might work (Fig. 8) pretty well. I look forward to the authors applying this technique to real data, and I wonder how observational uncertainty will impact the results given the size of the existing errors.

We agree that the application of the improved approximation method to real measurements seems promising. The present manuscript should be seen as a first step into that direction by assessing the potential of the methods within the "model world". The application of the methods laid out in this paper to real observational data will be the focus of the future work.

4. As an aside, the authors mentioned a number of times that their analysis won't work in the polar regions, yet they show these regions in the figures which is distracting. Perhaps cutting the figures at $\pm 50°$ might be reasonable.

Thank you for this comment. We certainly discussed and considered this comment. Although the reconstruction methods does not work at SH polar regions (because of dehydration by sedimentationing ice crystals), we think it is important to keep these regions in the plots. On the one hand this makes the problem evident to all readers directly at first glance. Also, cutting figures only at the SH would lead to a non-centered equator,

what could potentially confuse readers, as there is little dehydration in the polar Arctic stratosphere in winter in the NH.

---

## Referee Report (RR1)

Reviewer (Comments):
**Review of "Estimating Brewer-Dobson circulation trends from changes in stratospheric water vapour and methane" by Liubov Poshyvailo-Strube et al.**

**Recommendation: Publication after minor revision**

The paper is very well organised and written. The topic discussed here – quantifying the uncertainties in mean age of air (AoA) trend estimation from stratospheric water vapour and methane observations (for different assumptions) – is of high relevance. AoA is maybe still the most important proxy for estimating Brewer-Dobson circulation (BDC) trends from observations. Understanding BDC changes is still of high interest, because the BDC controls the distribution of trace gases (and also aerosols to some extend) in the stratosphere and therefore affects radiation (especially by changes of water vapour and ozone in the tropopause region). How BDC changes under changing climate conditions and how BDC changes impacts climate (the feedback mechanisms) are still not well known yet. This work advertise the use of a practical method for obtaining more reliable AoA trends from $H_2O$ and $CH_4$ observations. This is valuable for two reasons. First, long time series from satellites observations exist (and will be continued hopefully) for both tracers and second, the uncertainties and limitations of the method(s) are very thoroughly analysed here – a general prerequisite for trend analysis from observations.

However, some open questions remain and some points should be clarified/changed/added. The paper should be submitted after addressing the comments below.

**General comments:**

I highly appreciate the great effort and the way how the authors build up their argumentation, why it is reasonable and valuable using water vapour and methane observations to derive AoA trends. The argumentation based on the same general approach that Fritsch et al. (2020) used for analysing the sensitivity of AoA trends to the derivation method for non-linear increasing inert $SF_6$ – carrying out proof-of-concept of the method(s) applied to observations within the closed (or self-consistent) "model world" of CLaMS or EMAC, respectively.
As outlined in my recommendation, it is still an open and pressing question to diagnose BDC changes and I totally agree in general with the conclusions of this paper.

There are two general comments from my side:

1.) Total hydrogen in the stratosphere is defined here as the sum of $H_2O_{entry}$ + 2·$CH_4$. This definition is often used, but it might make sense to include also hydrogen ($H_2$) in this budget or at least shortly discuss the role of hydrogen for the stratospheric water vapour trend, especially in respect of a future hydrogen economy. Disregarding stratospheric moistening by increasing tropospheric hydrogen could lead to a misinterpretation of BDC trends in the future using only the conservation of total hydrogen as defined here.

2.) It is possible to deduce stratospheric circulation trends from FRF trends, but these FRF trends are generally a consequence of changing transit times (age spectra) and circulation patterns (pathways or path spectra). This means that unambiguous deduction of AoA trends from FRF trends is only valid under the assumption that only transit times (and not circulation patterns) change or that the chemical decay of a tracer is path-independent or that the changes in the circulation patterns compensate each other (for a specific non-inert tracer).

The path-dependency of FRF trends is to my opinion no problem in this paper. The reasons are, that you implicitly account for it by using AoA-FRF correlations for all methods (non-stationary or stationary) and that your conclusions are not affected: 1.) AoA trends from methane and water vapour are significantly affected by the assumed approximations and 2.) Using an idealised age spectra to calculate $H_2O_{|entry}$ and $CH_{4|entry}$ improves the APPROX method in respect of a more reliable AoA trend estimation.

However, it should be clarified that the reconstructed quantities $\Delta H_2O$ and $\Delta AoA$, as they are derived here, includes also possible changes in transport pathways and that disentangling the effect of changing transit times and transit pathways on the BDC is still an unsolved issue for diagnosing BDC changes from observations.

Finally a suggestion:

It might be worth to think about changing the title of this very sound paper to a question: "Estimating Brewer-Dobson circulation trends from changes in stratospheric water vapour and methane?". To my point of view, the answer is yes, but one have to take into account the uncertainties and limitations that you elaborated in this work.

For my feeling, posing a question would better fit to the storyline of this paper and the answer, that the improved method to obtain more reliable AoA trends derived from water vapour and methane observation is currently one of the few promising ways to estimate BDC changes from observations, would strengthen the conclusion of this paper.

**Specific comments:**

L.70-73: "*They (Hegglin et al., 2014) showed that ... are related to an accelerating shallow branch ... and to a deceleration of the deep branch of the BDC ...*"
For completeness this sentence should be extended by the following:
"... as suggested by Engel et al. (2009) and shown by Bönisch et al. (2011) for the same period."

L.79-81: "*The strength of the chemical source of $H_2O$ ...*"
It is true that methane (and hydrogen) oxidation is related to AoA, but AoA is only a measure for the transit time but not the transit pathway dependency (this is of particular interest, if you're a looking for changes in the BDC patterns). This differentiation should be added for clarity.

L.92-93: "*Precisely, the source region covers the potential temperature layer from 10 K below to 10 K above the WMO (lapse rate) tropopause.*"
Is this criteria sufficient in the Subtropical Jet (STJ) regions with strong distortion of the tropopause and even double tropopauses? If not, does it matter for this study?

L168-170: "*Trends in AoA... by using the conservation property of total hydrogen in the stratosphere, namely that the sum of $H_2O$ and two times $CH_4$ mixing ratios...*"
How about hydrogen (see general comments)?

L.185-186: "*The FRF is strongly affected by the vertical transport of the BDC. Hence, information on circulation trends (in particular on AoA) can be deduced from trends in FRF (Hegglin et al., 2014).*"
This is generally only valid for AoA trends under the assumption that only transit times and not circulation patterns change or that the chemical decay of a tracer is path-independent or if the changes in the circulation patterns compensate each other (see general comment 2.)).

L.189, EQ (6): It might be good to point out here, that the first two terms in the equation dependent only on changes in transit times and the third term (including Δα or ΔFRF) depends also on changing transit pathways (see general comment 2.)).

L.196: "*… and can be converted to an AoA trend.*"
See again general comment 2.).

L.232-234: "*The location of entry to the stratosphere is approximated as the 390-400 K layer between 30◦S-30◦N, which is located just above the cold point tropopause*"
Could this have an impact on your results, because the distance in potential temperature between the 390-400K level and the thermal (or dynamical) tropopause could be large, especially in the winter hemisphere at the edge of the defined entry region (>25°) (see also comment for L.92-93 above)?

L.259-261: "*Outside of the Southern high-latitude regions, the overall differences shown in Fig. 3c are small …*"
I would add here: "*Outside of the Southern high-latitude regions **and in the proximity of the extratropical tropopause (>30°N/S)**, the overall differences…*"
The reason for this is that direct in-mixing into the LMS not via the defined entry layer occurs and that especially water vapour mixing ratios at the extratropical tropopause are rather different from the mixing ratios at the tropical tropopause.

L.279-280: "*And, for instance, at the same FRF level of 0.3, the air at the Northern tropics (30◦ N-40◦N) is younger than at the Southern tropics (30◦S-40◦S) by almost half a year.*"
Is there a (simple) explanation for this?
This is an interesting finding and maybe it is worth to discuss it (or speculate about it).

L.294: "*… (e.g., Schoeberl et al., 2000, 2005; Ehhalt et al., 2007; Hegglin et al., 2014).*"
Would you please add Fritsch et al. (2020) here, because this is in my opinion also a highly relevant work on the topic – limitations (and improvements) for deriving AoA from real-world age tracers.

L.335: "*Thus, the accuracy of the estimated AoA changes from APPROX largely depends on the considered period.*"
It is true that AoA trend depends on the period, but it is likely that the main criteria is that the period has to be long enough to cover the internal variability. If the period is too short there will be random results for different sub-periods.

L.357-358: "*Hence, the good performance of the FULL method can be related to the fact that stratospheric entry $H_2O$ mixing ratios are not influencing the calculation.*"
To avoid misunderstandings, I would add here:
"… provided that some regions are excluded (as explained in section 3.1)."

**References:**

Bönisch, H., Engel, A., Birner, T., Hoor, P., Tarasick, D. W., and Ray, E. A.: On the structural changes in the Brewer-Dobson circulation after 2000, Atmos. Chem. Phys., 11, 3937-3948, 10.5194/acp-11-3937-2011, 2011.

---

## Author Response (AR2)

**Reply to the reviewer #3**
**Estimating Brewer-Dobson circulation trends from changes in stratospheric water vapour and methane**

We thank the referee for the detailed review and for the helpful comments and suggestions. We give a point-by-point reply below, where the reviewer comments are repeated in black. The replies to the reviewer's comments are in blue. The revised text is given in italics and in quotation marks, with the positions of the corrected sentences in the revised version noted in brackets.

**General remarks**

The paper is very well organised and written. The topic discussed here – quantifying the uncertainties in mean age of air (AoA) trend estimation from stratospheric water vapour and methane observations (for different assumptions) – is of high relevance... This work advertise the use of a practical method for obtaining more reliable AoA trends from H2O and CH4 observations. This is valuable... However, some open questions remain and some points should be clarified/changed/added. The paper should be submitted after addressing the comments below.

We thank the reviewer for this very encouraging comment and the suggestions for improvement. In the revised version, all suggested comments have been taken into account.

**General comments**

I highly appreciate the great effort and the way how the authors build up their argumentation, why it is reasonable and valuable using water vapour and methane observations to derive AoA trends... As outlined in my recommendation, it is still an open and pressing question to diagnose BDC changes and I totally agree in general with the conclusions of this paper. There are two general comments from my side.

1. Total hydrogen in the stratosphere is defined here as the sum of $H_2O_{entry}$ + 2·$CH_4$. This definition is often used, but it might make sense to include also hydrogen ($H_2$) in this budget or at least shortly discuss the role of hydrogen for the stratospheric water vapour trend, especially in respect of a future hydrogen economy. Disregarding stratospheric moistening by increasing tropospheric hydrogen could lead to a misinterpretation of BDC trends in the future using only the conservation of total hydrogen as defined here.

   Thank you for this important comment.

There is already a discussion about $H_2O$ sources in the introduction of the paper, and the limitations are mentioned with respect to $CH_4$ oxidation as the only considered source of stratospheric $H_2O$ (p3, L76).

However, following the reviewer's suggestions, we expanded the discussion on the causes for $H_2O$ changes in Sect. 2.3, see our main additions below. For further details, please refer to the difference .pdf file between the submitted and updated version of the manuscript.

- (p.6, L174) *"Changes in stratospheric $H_2O$ are determined by the stratospheric $H_2O$ entry mixing ratio through troposphere–stratosphere exchange (Fueglistaler and Haynes, 2005), and by chemical sources, mainly oxidation of $CH_4$ and molecular hydrogen ($H_2$) in the middle and high stratosphere (Dessler et al., 1994; Harries, 2015). $H_2O$ in the troposphere is continuously supplied from the Earth's surface. $CH_4$ is largely emitted at the Earth's surface because of anaerobic reactions, and $H_2$ is originated from biomass burning and other natural sources; $CH_4$ and $H_2$ are transported from the troposphere into the stratosphere. Based on satellite and balloon observations, the sum of the principal components of the hydrogen budget ($H_2O$, $2\times CH_4$ and $H_2$) is constant with altitude over most of the stratosphere (e.g., Dessler et al., 1994)."*

- (p.6, L182) *"... assuming that $H_2$ production from $CH_4$ oxidation is balanced by $H_2$ oxidation, namely that the sum of $H_2O$ and two times $CH_4$ mixing ratios is approximately constant..."*

- (p.7, L189) *"Note that the usage of the simple parameterization (see Eq. 4) for the ratio between oxidized $CH_4$ and produced $H_2O$ has its limitations, e.g., a ratio of 2 overestimates the production of $H_2O$ in the lower stratosphere and somewhat underestimates it in the upper stratosphere (Frank et al., 2018). It is questionable whether this parameterisation parameterisation can be used for future climate projections, when the BDC is expected to accelerate (e.g., Austin and Li, 2006; Li et al., 2008; Garcia and Randel, 2008), and, as a result, the transport of $H_2$ molecules becomes an important factor for the vertical profile of the $H_2O$ in the stratosphere. We also note that an increase in tropospheric $H_2$ might gain importance in a future hydrogen economy (e.g., Vogel et al., 2012)."*

2. It is possible to deduce stratospheric circulation trends from FRF trends, but these FRF trends are generally a consequence of changing transit times (age spectra) and circulation patterns (pathways or path spectra). This means that unambiguous deduction of AoA trends from FRF trends is only valid under the assumption that only transit times (and not circulation patterns) change or that the chemical decay of a tracer is path-independent or that the changes in the circulation patterns compensate each other (for a specific non-inert tracer).
The path-dependency of FRF trends is to my opinion no problem in this paper. The reasons are, that you implicitly account for it by using AoA-FRF correlations for all methods (non-stationary or stationary) and that your conclusions are not affected: 1.) AoA trends

from methane and water vapour are significantly affected by the assumed approximations and 2.) Using an idealised age spectra to calculate $H_2O_{entry}$ and $CH_{4entry}$ improves the APPROX method in respect of a more reliable AoA trend estimation.

However, it should be clarified that the reconstructed quantities $\Delta H2O$ and $\Delta AoA$, as they are derived here, includes also possible changes in transport pathways and that disentangling the effect of changing transit times and transit pathways on the BDC is still an unsolved issue for diagnosing BDC changes from observations.

*Thank you for this comment. This point was already partially addressed in the introduction of the previous paper version, moreover, some clarifications have been added there (p.3, L81).*
*"The strength of the chemical source of $H_2O$ depends on transit path of air since entering the stratosphere and transit time, and is thus related to AoA, which in turn is a measure for only the transit time but not the transit pathways dependency. The full complexity of these processes is very challenging to represent in the analysis of stratospheric $H_2O$, in particular, it remains an issue to disentangle the effects of changing transit time and changing transport pathways when diagnosing trends of the BDC from observations."*

3. It might be worth to think about changing the title of this very sound paper to a question... For my feeling, posing a question would better fit to the storyline of this paper and the answer, that the improved method to obtain more reliable AoA trends derived from water vapour and methane observation is currently one of the few promising ways to estimate BDC changes from observations, would strengthen the conclusion of this paper.

*Thank you for this suggestion. We formulated the title of the paper as a question. The current title is "How can Brewer-Dobson circulation trends be estimated from changes in stratospheric water vapour and methane?"*

**Specific comments**

- L.70-73: *"They (Hegglin et al., 2014) showed that... are related to an accelerating shallow branch ... and to a deceleration of the deep branch of the BDC..."*
  For completeness this sentence should be extended by the following:
  "... as suggested by Engel et al. (2009) and shown by Bönisch et al. (2011) for the same period."

  *Thank you for the remark. The suggested text was added in the revised version (p3, L70).*
  *"Bönisch et al. (2011) and Hegglin et al. (2014) showed that a decrease in the $H_2O$ mixing ratios in the lower stratosphere, below about $10\,hPa$, and an increase in the $H_2O$ mixing ratios above this level from the mid 1980s to 2010 are related to an accelerating shallow branch of the BDC (decreasing AoA below about $10\,hPa$) and to a decelerating deep branch of the BDC (increasing AoA above), as originally suggested by Engel et al. (2009)."*

- L.79-81: *"The strength of the chemical source of H2O..."*
  It is true that methane (and hydrogen) oxidation is related to AoA, but AoA is only a measure for the transit time but not the transit pathway dependency (this is of particular interest, if you're a looking for changes in the BDC patterns). This differentiation should be added for clarity.

  Thank you for the specific comment. We have added the proposed statement to the text (p3, L81).
  *"The strength of the chemical source of $H_2O$ depends on the transit time and the transit path of air since entering the stratosphere and, thus, is related to AoA, which in turn is a measure for only the transit time but not the transit pathways dependency."*

- L.92-93: *"Precisely, the source region covers the potential temperature layer from 10 K below to 10 K above the WMO (lapse rate) tropopause."*
  Is this criteria sufficient in the Subtropical Jet (STJ) regions with strong distortion of the tropopause and even double tropopauses? If not, does it matter for this study?

  We agree that the choice of the source region causes some uncertainty in the analysis as it does not exactly match the region defining the $H_2O$ and $CH_4$ entry. We added the related sentence at the end of Sect. 2.2 (p6, L163).
  *"As a remark, this specific choice of the source region causes the uncertainty in our analyses as it does not exactly correspond to the region defining the $H_2O$ and $CH_4$ entry mixing ratios; the tropically controlled transition region bounds between approximately 380 K and 450 K (Rosenlof et al., 1997; Li et al., 2012). However, this mismatch impacts the results only close to the tropopause, so the reconstruction of $H_2O$ and $CH_4$ by the modelled age spectrum ensures the reliability of the method in most of the stratosphere."*

- L168-170: *"Trends in AoA... by using the conservation property of total hydrogen in the stratosphere, namely that the sum of H2O and two times CH4 mixing ratios..."*
  How about hydrogen (see general comments)?

  Thank you for the comment. We have added the statement about hydrogen in the revised version (p6, L182).
  *"Trends in AoA can be calculated from trends in stratospheric $H_2O$ mixing ratios by using the conservation property of total hydrogen in the stratosphere and assuming that $H_2$ production from $CH_4$ oxidation is balanced by $H_2$ oxidation, namely that the sum of $H_2O$ and two times $CH_4$ mixing ratios is approximately constant..."*

- L.185-186: *"The FRF is strongly affected by the vertical transport of the BDC. Hence, information on circulation trends (in particular on AoA) can be deduced from trends in FRF (Hegglin et al., 2014)."*
  This is generally only valid for AoA trends under the assumption that only transit times and not circulation patterns change or that the chemical decay of a tracer is path-independent or if the changes in the circulation patterns compensate each other (see general comment 2.)).

We inserted the required clarification to the text (p7, L207).
*"It should be noted that the change in FRF is due to transit times (age spectra) and circulation pathways (path spectra) changes, but AoA is a measure for only transit times and not the transit pathways dependency."*

- L.189, EQ (6): It might be good to point out here, that the first two terms in the equation dependent only on changes in transit times and the third term (including $\Delta\alpha$ or $\Delta$FRF) depends also on changing transit pathways (see general comment 2.)).

Thank you for the remark, we added the mentioned clarification to the text (p7, L214).
*"Note that $\Delta H_2O_{[entry]}(r,\ t)$ and $\Delta CH_{4[entry]}(r,\ t)$ depend only on changes in transit times, while $\Delta\alpha(r,\ t)$ generally also depends on the changes in transport pathways."*

- L.196: *"... and can be converted to an AoA trend."*
See again general comment 2.).

We inserted an additional sentence to the revised version of the paper (p8, L221).
*"Besides, the dependency of FRF changes on circulation pathways is implicitly taken into account by the AoA-FRF correlation functions in all used methods (Sect. 2.4)."*

- L232-234: *"The location of entry to the stratosphere is approximated as the 390-400K layer between 30°S-30°N, which is located just above the cold point tropopause."*
Could this have an impact on your results, because the distance in potential temperature between the 390-400K level and the thermal (or dynamical) tropopause could be large, especially in the winter hemisphere at the edge of the defined entry region (>25°) (see also comment for L.92-93 above)?

Thank you for this question. We added the explanation to the revised version of the paper (p10, L261).
*"The small difference between the age spectrum source region (tropopause $\pm 10\,K$) and the trace gases entry region has only a negligible impact on our results from above $\approx 420\,K$ due to the small difference in the transit time between the two regions."*

- L.259-261: *"Outside of the Southern high-latitude regions, the overall differences shown in Fig. 3c are small ..."*
I would add here: *"Outside of the Southern high-latitude regions and in the proximity of the extratropical tropopause (>30°N/S), the overall differences..."*
The reason for this is that direct in-mixing into the LMS not via the defined entry layer occurs and that especially water vapour mixing ratios at the extratropical tropopause are rather different from the mixing ratios at the tropical tropopause.

Thank you for the remark. The suggested text was added in the revised version (p11, L287).
*"Outside of the Southern high-latitude regions and in the proximity of the extratropical tropopause (>30°N/S), the overall differences shown in..."*

- L.279-280: *"And, for instance, at the same FRF level of 0.3, the air at the Northern tropics (30°N-40°N) is younger than at the Southern tropics (30°S-40°S) by almost half a*

*year."*
Is there a (simple) explanation for this? This is an interesting finding and maybe it is worth to discuss it (or speculate about it).

This is likely related to the hemispheric asymmetry in the BDC, which is stronger in the NH during boreal winter; the figure you are referring to is shown for January, 2000. We have added clarifications to the text (p12, L308).
*"It is likely due to stronger and deeper BDC in the Northern hemisphere during boreal winter (e.g., Rosenlof, 1995; Butchart, 2014) causing air parcels of the same age to travel deeper pathways through the stratosphere and experience more chemical depletion compared to the Southern hemisphere."*

- L.294: *"...(e.g., Schoeberl et al., 2000, 2005; Ehhalt et al., 2007; Hegglin et al., 2014)."*
Would you please add Fritsch et al. (2020) here, because this is in my opinion also a highly relevant work on the topic – limitations (and improvements) for deriving AoA from real-world age tracers.

The suggested reference was added in the revised version (p14, L325).
*"... about the age spectrum and its shape (e.g., Schoeberl et al., 2000, 2005; Ehhalt et al., 2007; Hegglin et al., 2014; Fritsch et al., 2020)."*

- L.335: *"Thus, the accuracy of the estimated AoA changes from APPROX largely depends on the considered period."*
It is true that AoA trend depends on the period, but it is likely that the main criteria is that the period has to be long enough to cover the internal variability. If the period is too short there will be random results for different sub-periods.

Thank you for the remark. We rephrased the sentence as suggested (p15, L365).
*"Thus, the accuracy of the estimated AoA changes from APPROX largely depends on the considered period, which should be long enough to ensure that the effects of variability is small."*

- L.357-358: *"Hence, the good performance of the FULL method can be related to the fact that stratospheric entry H2O mixing ratios are not influencing the calculation."*
To avoid misunderstandings, I would add here: *"... provided that some regions are excluded (as explained in section 3.1)."*

Thank you for the remark. We have added the suggested text in the revised version of the paper (p16, L387).
*"Hence, the good performance of the FULL method can be related to the fact that stratospheric entry $H_2O$ mixing ratios are not influencing the calculation, provided that the polar regions are excluded (as explained in Sec. 3.1)."*